# CONSISTENCY CHECKS FOR LANGUAGE MODEL FORECASTERS

**Daniel Paleka** [*]
ETH Zurich

**Abhimanyu Pallavi Sudhir** [*]
University of Warwick

**Alejandro Alvarez**
Independent

**Vineeth Bhat**
IIIT Hyderabad

**Adam Shen**
Columbia University

**Evan Wang**
Cornell University

**Florian Tramèr**
ETH Zurich

## ABSTRACT

Forecasting is a task that is difficult to evaluate: the ground truth can only be known in the future. Recent work showing LLM forecasters rapidly approaching human-level performance begs the question: how can we benchmark and evaluate these forecasters *instantaneously*? Following the consistency check framework, we measure the performance of forecasters in terms of the consistency of their predictions on different logically-related questions. We propose a new, general consistency metric based on *arbitrage*: for example, if a forecasting AI illogically predicts that both the Democratic and Republican parties have 60% probability of winning the 2024 US presidential election, an arbitrageur can trade against the forecaster's predictions and make a profit. We build an automated evaluation system that generates a set of base questions, instantiates consistency checks from these questions, elicits the predictions of the forecaster, and measures the consistency of the predictions. We then build a standard, proper-scoring-rule forecasting benchmark, and show that our (instantaneous) consistency metrics correlate with LLM forecasters' ground truth Brier scores (which are only known in the future). We also release a consistency benchmark that resolves in 2028, providing a long-term evaluation tool for forecasting.

## 1 INTRODUCTION

*Prediction markets* are markets that pay out contingent on an event. For a market such as "$1 if Jeb Bush is elected President in 2028", the price reflects the "market estimate" for the probability of that event. Prediction markets are a promising tool for aggregating information from disparate sources to arrive at the most correct possible belief after taking into account all relevant information (Arrow et al., 2008; Hanson, 2002).

Until 2024, LLM forecasters generally performed poorly relative to human forecasters (Zou et al., 2022b; Schoenegger and Park, 2023). However, recent works (Halawi et al., 2024; Schoenegger et al., 2024; Phan et al., 2024) suggest that LLM-based forecasters can rival human forecasts on forecasting websites such as Metaculus, PredictIt, and Manifold Markets.

A key question emerges: *once LLM forecasters are better than human ones, how can we efficiently evaluate their predictions?* In particular, long-term forecasting questions are very important for decision-making (Tetlock et al., 2024; Muehlhauser, Luke, 2019), and finding ground truth for evaluation in such contexts is infeasible by virtue of the questions resolving far in the future.

One approach, proposed by Fluri et al. (2023), is that even when we cannot evaluate the *correctness* of LLM decisions, we can evaluate their *logical consistency*. For example, if an LLM forecaster gives probabilities 0.5 and 0.7 to "Will Trump be elected US president?" and "Will someone other than Trump be elected US president?", this is necessarily inconsistent. Fluri et al. (2023) demonstrated that GPT-4 and GPT-3.5-turbo, when asked one-sentence forecasting questions, were inconsistent on simple logical checks such as negation.

---

[*]Equal contribution. Corresponding: `daniel.paleka@inf.ethz.ch`. Author contributions in *Contributions*.

Our contributions in this work are as follows:

**1) Principled metrics for consistency.** In Section 2, we introduce a theoretical framework for measuring consistency violations of binary forecasts, based on two metrics: an *arbitrage metric*, based on market arbitrage, and a *frequentist metric*, based on hypothesis testing. We apply these metrics to 10 different logical consistency rules (see Table 3): NEGATION, PARAPHRASE, CONSEQUENCE, ANDOR, AND, OR, BUT, COND, CONDCOND and EXPEVIDENCE.

**2) A consistency evaluation pipeline for binary forecasters.** In Section 3, we introduce a *consistency evaluation pipeline* for LLM forecasters. We create two forecasting datasets with known ground truth resolutions: one scraped from prediction markets, and one synthetically generated from news articles. Both datasets include only events that happen past the training data cutoff of all forecasters we test, and resolve before September 2024. We then generate tuples of forecasting questions satisfying logical consistency rules with associated consistency metrics.

**3) Consistency correlates with ground truth forecasting performance.** Our consistency metrics are novel performance metrics for forecasters that can be computed right away, no matter the time horizon. Of course, forecasters could also be evaluated using *backtesting*, asking past questions with known ground truth resolutions. Yet, backtesting LLM forecasters can be challenging if we do not have clear information about the models' training data contents. Moreover, there may be new types of questions that we want to evaluate forecasters on, for which we do not have appropriate past results (e.g., questions related to pandemics before 2020). It is thus natural to ask: *can consistency metrics tell us anything about future forecasting performance?*

In Section 4, we show that for all forecasters we test, our consistency metrics correlate positively with forecasting performance (as measured by the Brier score) on both our benchmark datasets. The correlation varies across consistency checks, with some logical checks (e.g., consistency of conditional probabilities) having over $R = 0.9$ correlation with forecasting performance, while other logical tests provide little signal. We hypothesise that this analysis can extend to smarter forecasters and longer time horizons, to provide instantaneous feedback on forecaster performance.

**4) Scaling inference-time compute can improve consistency for some logical checks, but fails to generalize.** Since we find that consistency correlates with forecasting performance, it is natural to ask whether we can improve forecasters by making them more consistent. Unfortunately, we find that natural ways of improving consistency tend to overfit to specific consistency checks and do not generalize.

Specifically, we design `ArbitrageForecaster`: a forecaster that "patches" some base forecaster's output by generating logically related questions and "arbitraging" the base forecaster's forecasts for these related questions against each other. In Section 5 and Appendix G, we show that `ArbitrageForecaster` improves consistency on checks that we optimize against, but this improvement does not generalize to other held-out consistency checks, nor does it improve the actual forecasting performance.

**5) A long-horizon forecasting consistency benchmark.** We create a long-horizon benchmark of 3,000 consistency checks for forecasts resolving in 2028. Our benchmark spans questions on various topics for which we will have no ground truth for more than three years, and thus serves as a nice testing ground for advanced LLM forecasters.

We release the full code [1] and the datasets [2] used in the paper.

## 2 A THEORETICAL FRAMEWORK FOR FORECASTING CONSISTENCY

*Notation.* Let $\mathrm{Prop}$ denote the set of forecasting questions we are interested in, $\Theta$ denote the set of possible outcomes/resolutions for an individual questions. In this paper, we focus on $\mathrm{Prop}$ as a set of binary forecasting questions, so $\Theta = \{\top, \bot\}$. A *Forecaster* is then a map $\mathbb{F} : \mathrm{Prop} \to [0, 1]$. One special forecaster is the ground truth resolutions $\theta : \mathrm{Prop} \to \Theta$, returning 1 and 0 probability for $\{\top, \bot\}$, respectively.

---

[1] https://github.com/dpaleka/consistency-forecasting
[2] https://huggingface.co/datasets/dpaleka/ccflmf

For conditional questions that can resolve to None, we also have optional resolutions $\Theta' := \Theta \cup \{\text{None}\} = \{\top, \bot, \text{None}\}$. We focus on binary questions following Halawi et al. (2024). Our methods can in principle be extended to study consistency between general probability distributions in forecasting, such as the ones discussed in Gooen (2024).

## 2.1 Consistency checks and inconsistency metrics

In line with Fluri et al. (2023), a consistency check is conceptualized as a pair of n-ary relations: $\mathcal{R} : \text{Prop}^n \to \{\top, \bot\}$ in question space, $\mathcal{S} : [0, 1]^n \to \{\top, \bot\}$ in forecast space, and a predicate for $\mathbb{F}$ such that $\mathcal{R}(x_1, \ldots x_n) \implies \mathcal{S}(\mathbb{F}(x_1), \ldots \mathbb{F}(x_n))$. In particular, this assertion must be satisfied by all feasible $\theta$, and also any "correct" forecasts generated by a world model that accurately accounts for aleatoric uncertainty. Violation of consistency is measured by some violation metric $\mathcal{V} : [0, 1]^n \to \mathbb{R}$ which must satisfy $\mathcal{V}(\mathbb{F}(x_1), \ldots \mathbb{F}(x_n)) = 0 \iff \mathcal{S}(\mathbb{F}(x_1), \ldots \mathbb{F}(x_n))$. For example, intuitively, the "negation" check NEGATION is given by the relation $\mathcal{R}(x_1, x_2) := x_1 = \neg x_2$ on questions, and the relation $\mathcal{S}(\mathbb{F}(x_1), \mathbb{F}(x_2)) := \mathbb{F}(x_1) + \mathbb{F}(x_2) \approx 1$ on forecasts. The full table of the consistency checks we use is given in Appendix B.

Improving upon Fluri et al. (2023), we derive $\mathcal{V}$ from $\mathcal{R}$ in a principled way, handling all types of logical consistency checks simultaneously. We introduce two new *inconsistency metrics*: the *arbitrage metric* and the *frequentist metric* for measuring logical inconsistency in probabilistic forecasts.

### 2.1.1 Arbitrage metric

The arbitrage metric is conceptualized as the minimum profit that an arbitrageur can be guaranteed making bets against the forecaster's predictions. More precisely: suppose that the forecaster's probabilities $\mathbb{F}(x_1), \ldots \mathbb{F}(x_n)$ were prices offered by a logarithmic market maker [3] with market subsidy parameter \$1. If these probabilities are inconsistent, then there are prices $p_1, \ldots p_n$ that an arbitrageur could bring to the market such that it is guaranteed to make a profit against the market-maker, no matter the outcome of each question. We define $\mathcal{V}(\mathbb{F}(x_1), \ldots \mathbb{F}(x_n))$ as the maximum achievable "minimum profit" that the arbitrageur can guarantee by choosing appropriate $p_1, \ldots p_n$. We further denote by $\mathcal{A}(\mathbb{F}(x_1), \ldots \mathbb{F}(x_n))$ the set of prices $p_1, \ldots p_n$ that maximize the minimum profit:

$$(\arg\max, \max)_{p \in [0,1]^n} \min_{\omega \in \Omega} \sum_{i=1}^{n} \left(\log p_i - \log \mathbb{F}(x_i)\right) \delta_{\omega(i)=\top} + \left(\log\left(1 - p_i\right) - \log\left(1 - \mathbb{F}(x_i)\right)\right) \delta_{\omega(i)=\bot}$$

(1)

Here $\Omega := \{\omega \in \Theta'^n \mid \mathcal{R}(\omega)\}$ is the set of all possible consistent resolutions of this tuple. A more general version of 1 is given in Appendix D, along with specific worked-out examples of the arbitrage metric for each consistency check, and details on how we compute it; as an example, the arbitrage metric for the Negation Check can be derived exactly (Appendix D.2):

$$\mathcal{V}(\mathbb{F}(x), \mathbb{F}(\neg x)) = -2 \log \left( \sqrt{\mathbb{F}(x)(1 - \mathbb{F}(\neg x))} + \sqrt{(1 - \mathbb{F}(x))\mathbb{F}(\neg x)} \right)$$

To illustrate: $\mathcal{V}(0.5, 0.6) \approx 0.01$, $\mathcal{V}(0.5, 0.51) \approx 10^{-4}$. The metric is more sensitive to violations for probabilities very close to 0 or 1, due to the logarithmic market maker. In our evals, for all types of checks, we say that a sampled check does not pass if $\mathcal{V} \geq 0.01$. We have to pick some hyperparameter as an inconsistency threshold; we set it to correspond to giving 110% probability in total to the events of Republican and Democratic parties winning the US presidential election.

### 2.1.2 Frequentist metric

We also compute a different, *frequentist* consistency metric. Consider a Monte Carlo forecaster that samples a world model $n$ times, and for any event, returns the fraction of samples in which the event occurs. The frequentist metric is the number of standard deviations a given tuple forecast is off from

---

[3] A *logarithmic market maker* with subsidy $w$ is a market maker who adjusts prices in response to trades such that the trader's reward for moving the probability of a true-resolving sentence from $p_0$ to $p'$ is $w \log p' - w \log p_0$. For further background on scoring rules and the associated market makers, see Appendix D, Berg and Proebsting (2009), or Hanson (2002).

the mean Monte Carlo forecast, scaled to be independent of $n$. We say that a consistency violation happened if the number of standard deviations away from the mean of the null is at least as in the $(0.5, 0.6)$ case described in Section 2.1.1. The full description is given in Appendix E.

### 2.1.3 INTUITION ON CONSISTENCY METRICS

Our metrics address two major obstacles with measuring inconsistency: *tolerance to noise* and *principled aggregation of inconsistency scores*.

**Tolerance to noise.** In the standard Bayesian setting, beliefs are either consistent or not: there either is a Dutch book (a way to bet against the forecaster's beliefs to get infinite profit) or the probabilities are perfectly consistent. In practice, forecasters' beliefs (even on the same question) are never perfectly consistent across runs. If an election model has a presidential candidate at 48% with one random seed and 50% on the other, this is not a reason to discard it as completely flawed. Hence, instead of being a binary measure of consistency, our metrics increase smoothly with inconsistency.

**Principled aggregation and comparison of inconsistency scores** Fluri et al. (2023) developed a set of inconsistency checks, used an ad hoc metric for each check they used, and normalized the scores to [0, 1]. There are two important issues with their approach:

1. The metrics in their work are mostly *linear* and would treat the inconsistencies of $(0.5, 0.6)$ and $(0.89, 0.01)$ on the NEGATION check as equally bad, which is counterintuitive in many applications.

2. It is unclear how to compare and aggregate scores from different consistency checks.

Our approach ensures that all consistency scores share a common "unit". For example, in the arbitrage metric, to aggregate inconsistencies, we sum up the profit made by an arbitrageur across questions.

## 3 PIPELINE OVERVIEW

We illustrate the steps in our data collection pipeline below, and provide more details on each individual steps:

$$\boxed{\textit{Online platforms, news, topics}} \xrightarrow[\text{+scraping}]{\text{synthetic}} P \xrightarrow[\text{instantiation}]{\text{tuple}} (P, Q) \xrightarrow{\mathbb{F}} (p, q) \xrightarrow{\mathcal{V}} \mathcal{V}(p, q)$$

- $(\cdots \longrightarrow P)$ We first prepare datasets of **base questions** in multiple ways:

    (a) Scraping questions from online platforms such as Manifold and Metaculus;

    (b) A ground-truth resolved dataset synthetically generated from news articles;

    (c) Synthetic generation on questions on a list of topics such as Politics, Science, Economics, etc.

    For the first two of the above, we also include the *ground truth resolution* for each question. We discuss all of these in more detail in Section 3.1.

- $(P \longrightarrow (P, Q))$ The base questions are synthetically **instantiated into tuples** that must satisfy certain consistency checks. For example, every single base question $P$ is instantiated into a tuple $(P, \neg P)$; and pairs of mutually relevant base questions $P, Q$ are instantiated into tuples like $(P, Q, P \wedge Q, P \vee Q)$.

- $((P, Q) \xrightarrow{\mathbb{F}} (p, q))$ The forecaster is separately queried to elicit **forecasts** on each question, resulting in forecast tuples that should, if the forecaster is consistent, satisfy consistency properties. For example, for a size-two tuple where $Q = \neg P$, it should satisfy $p + q = 1$.

- $((p, q) \xrightarrow{\mathcal{V}} \mathcal{V}(p, q))$ We score each tuple of forecasts for consistency with both of our violation metrics.

Examples of data at each step of the pipeline are given in Appendix C. The prompts and LLM calls used in each step before forecasting are given in Appendix H.

### 3.1 GENERATING AND SCRAPING FORECASTING QUESTIONS

**Forecasting question format.**   Each forecasting question includes a title that states the main question, a body that provides detailed resolution criteria, and a resolution date, along with optional fields such as metadata and creation date.

**Real prediction market questions.**   We scrape questions from two forecasting platforms, Metaculus and Manifold Markets, and only use questions that both resolved and were initially set to resolve between May 1, 2024, and August 15, 2024. This leaves us with over 500 questions, of which 242 pass our verification step (see end of this subsection). An example of a processed question, including its relevant details, is provided in Appendix C.1.

**Generating forecasting questions from NewsAPI articles.**   To generate forecasting questions with known resolutions, we use articles sourced from NewsAPI. We focus on articles describing concrete events rather than opinion pieces. To mitigate biases towards positive resolutions (as most questions derived from an article would typically resolve to True), we employ reference class spanning - using an LLM to modify key entities in the questions while keeping the overall thematic structure intact. Each question's ground-truth resolution is verified using the Perplexity API with internet access, yielding ground truth resolution labels with less than a 5% error rate in our testing. We compile a total of 2,621 ground-truth resolved forecasting questions resolving between July 1, 2024, and August 21, 2024. Of these, we use a subset of 1,000 to test the relationship between consistency violation and accuracy. Further details regarding the pipeline can be found in Appendix K.

**Synthetic question generation.**   We generate questions by few-shot prompting, we sample six examples of forecasting questions, as style examples, as well as a set of tags (Brazil, NBA...) to diversify the generated questions. We generate question titles, deduplicate them using `text-embedding-3-small` embeddings from OpenAI, and then for each title we use `gpt-4o` to create the question body and resolution date. With this method we create 1,000 forecasting questions that resolve either by or in 2028. More details are in Appendix H.

**Verification and improvement from human feedback.**   In all of the above steps, we filter generated questions in using `gpt-4o` to check for properties such as the coherence between the body and title, the clarity and precision of the resolution criteria, and whether the question is about actual world events. Questions failing this step are discarded. To develop this step, we used a feedback form for human reviewers (authors of this paper) to suggest modifications to generated questions. These suggestions inform refinements to prompts and few-shot examples in our pipeline. An example of the feedback form is provided in Appendix I.

### 3.2 INSTANTIATING TUPLES OF QUESTIONS FOR CONSISTENCY CHECKS

The base forecasting questions are subsequently used to synthetically generate tuples of logically related questions. For example, a pair of base questions $(P, Q)$ can be used to generate a 4-tuple $(P, Q, P \land Q, P \lor Q)$ for ANDOR, or a 3-tuple $(P, \neg P \land Q, P \lor Q)$ for BUT (see Appendix B for details). The main question content (titles and bodies) were generated synthetically (using `gpt-4o`), while the resolution dates and other properties were calculated systematically (e.g. the `max` of the resolution dates of the base questions).

We then conduct two measures to ensure the instantiated tuples are correct and sensible: relevance scoring, and verification that the tuples of questions indeed describe logically related events.

**Relevance scoring.**   When combining base questions into tuples, we have to take care to avoid off-distribution questions like "Is SpaceX going to be worth \$200B by 2030, given that Sri Lanka's rice production grows 40% by 2040?". For tuples instantiated from more than one base question, we sort 2000 potential base question combinations by their "relevance score", obtained by querying an LLM and asking it to score how relevant the questions are to one another, and choose the top 200 for each consistency check. See Figure 15 for details.

**Verification.** The instantiated tuples of questions are then passed to another LLM call to reject if they do not fit their intended structure; for example, we detect if the resolution criteria of the second question are not truly a negation of the resolution criteria of the first question. Examples of verification prompts are given in Appendix H.

## 3.3 Eliciting forecasts

We test a range of forecasters based on various LLM models (`gpt-4o`, `gpt-4o-mini`, `claude-3.5-sonnet`, `llama-3.1-8B`, `llama-3.1-70B`, `llama-3.1-405B`, `o1-mini` and `o1-preview`) with and without chain-of-thought prompting: see Appendix F for details. We run each of these forecasters on 5000 tuples in total (for each of the 10 checks, we use 200 tuples from scraped questions and 300 from NewsAPI questions), except for `o1-preview`, which we test on 50 tuples per check only due to cost constraints. We could not test forecasters from Halawi et al. (2024) due to API deprecations; see Section 6.

## 4 Results

We evaluate a range of forecasters on the datasets described above, for both consistency and ground truth Brier score. We note that the Brier score as the standard metric of forecasting accuracy depends both on model capabilities and the training data cutoff: it should not be surprising for a stronger model to have a worse Brier score if its training data cutoff is earlier than for a weaker model. The full list of forecasters is in Appendix F.

For all data analysis in this section, we exclude forecasters that have Brier score worse than random guessing (0.25), such as the basic setup with `llama-3.1-8B`, as it would unfairly advantage our case of "correlating consistency with accuracy".

**Average consistency scores correlate strongly with forecasting performance.** We can aggregate the consistency scores across all checks for each forecaster by aggregating either the arbitrage or the frequentist violations.

We plot the average Brier score against the three aggregate consistency scores in Figure 1.

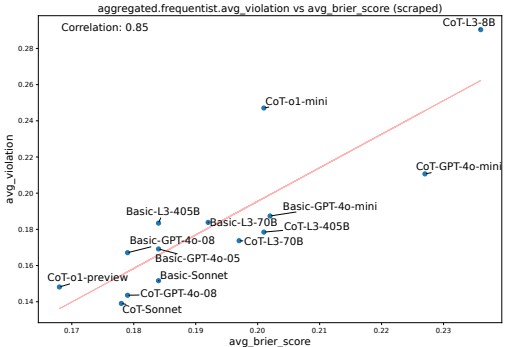
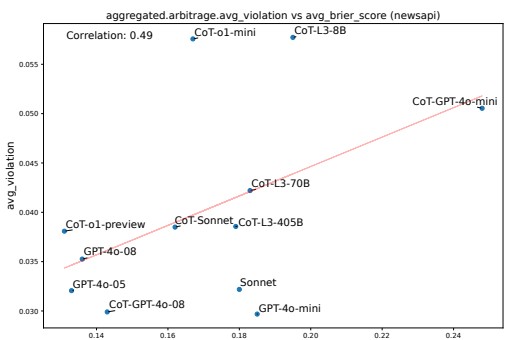

(a) Aggregate frequentist metric on the scraped forecasting question dataset.

(b) Aggregate arbitrage metric on the NewsAPI questions dataset.

Figure 1: Scatter plots showing the relationship between consistency metrics and average Brier scores for different forecasters. Each point represents a forecaster, with the x-axis showing the average Brier score and the y-axis showing the consistency metric. The y-axis values are aggregated across all checks for each forecaster and averaged over the instantiated consistency check tuples. Lower scores are better for both axes.

**Bayesian consistency checks are the best proxies for forecasting performance.** Figure 2a shows the strong correlation between certain consistency checks from Table 3 and average Brier scores across different forecasters. This relationship suggests that Cond, which tracks logical consistency in conditional probability estimates, serves as a proxy for overall forecasting accuracy, *without knowing how the questions resolved*.

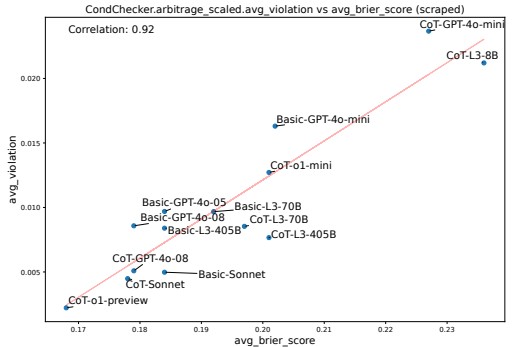 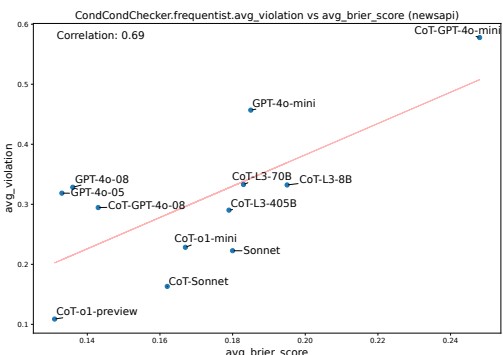

(a) COND arbitrage metric on the scraped forecasting question dataset.

(b) CONDCOND frequentist metric on the NewsAPI questions dataset.

Figure 2: Both COND and CONDCOND consistency metrics see Table 3 show a strong correlation with forecasting accuracy as measured by the Brier score.

**Certain consistency metrics are not well correlated with forecasting performance.** The measured correlations between the consistency checks and Brier scores are given in Table 1. We see that some checks yield higher signal on the ground truth performance than others. Aggregating different consistency metrics seems to improve the correlation.

We note that the selection of forecasters we test is quite limited, so we cannot guarantee the trends here are representative of future LLM forecasters. Part of the correlation can be attributed to better models being both more consistent and better forecasters. For comparison, the correlations of the Brier score of our forecasters and the MMLU Hendrycks et al. (2020) (college split) error rate on the best approximation of our forecasters in Appendix F are 0.38 and 0.55 on the NewsAPI and scraped datasets, respectively.

We include all data (questions, tuples, forecasts, and scores) in the supplementary material.

Table 1: Correlation of consistency metrics with Brier score, across both of our base question datasets and the derived consistency check tuples.

|  | Scraped | | NewsAPI | |
|---|---|---|---|---|
|  | Arbitrage | Frequentist | Arbitrage | Frequentist |
| NEGATION | 0.60 | 0.67 | -0.36 | -0.13 |
| PARAPHRASE | 0.57 | 0.61 | 0.13 | 0.24 |
| CONSEQUENCE | 0.51 | 0.52 | 0.21 | 0.30 |
| ANDOR | 0.20 | 0.25 | 0.02 | 0.06 |
| AND | 0.68 | 0.72 | 0.54 | 0.71 |
| OR | 0.14 | 0.24 | -0.24 | -0.31 |
| BUT | 0.20 | 0.67 | 0.63 | 0.77 |
| COND | 0.92 | 0.87 | 0.71 | 0.69 |
| CONDCOND | 0.78 | 0.71 | 0.75 | 0.69 |
| EXPEVIDENCE | 0.20 | 0.77 | -0.11 | 0.06 |
| Aggregated | 0.62 | 0.85 | 0.49 | 0.66 |

**Even good reasoning models are inconsistent.** We give the full set of consistency metrics for OpenAI's `o1-mini` in Table 2. The `Frac` column counts the fraction of tuples for which the violation exceeded a certain threshold; see the full exposition of what the thresholds mean in Appendices D and E. The frequentist metric is not directly comparable to the arbitrage metric, but the respective violation counts ("Frac" in the table) are.

OpenAI's `o1-mini` forecaster, despite being one of the best reasoning models so far, violates consistency checks more than the $(0.5, 0.6)$ threshold from Section 2 very often.

Table 2: Consistency metrics for o1-mini.

| | Scraped | | | | NewsAPI | | | |
| | Arbitrage | | Frequentist | | Arbitrage | | Frequentist | |
| Check | Avg | Frac | Avg | Frac | Avg | Frac | Avg | Frac |
|---|---|---|---|---|---|---|---|---|
| NEGATION | 0.07 | 58% | 0.26 | 61% | 0.08 | 52% | 0.27 | 56% |
| PARAPHRASE | 0.07 | 56% | 0.26 | 61% | 0.06 | 53% | 0.24 | 56% |
| CONSEQUENCE | 0.03 | 27% | 0.13 | 29% | 0.03 | 18% | 0.10 | 19% |
| ANDOR | 0.09 | 65% | 0.34 | 71% | 0.07 | 57% | 0.29 | 67% |
| AND | 0.02 | 24% | 0.11 | 27% | 0.03 | 23% | 0.11 | 24% |
| OR | 0.11 | 48% | 0.30 | 50% | 0.05 | 48% | 0.21 | 50% |
| BUT | 0.11 | 60% | 0.40 | 79% | 0.11 | 63% | 0.38 | 80% |
| COND | 0.04 | 41% | 0.22 | 52% | 0.07 | 66% | 0.29 | 70% |
| CONDCOND | 0.03 | 30% | 0.19 | 45% | 0.04 | 54% | 0.23 | 71% |
| EXPEVIDENCE | 0.04 | 47% | 0.27 | 69% | 0.05 | 45% | 0.28 | 63% |
| **Aggregated** | 0.06 | — | 0.25 | — | 0.06 | — | 0.24 | — |

**Long-horizon consistency benchmark.** The results of the previous section indicate that, even on longer time horizons where it's not possible to have ground truth resolutions, we can still evaluate and compare different forecasters via consistency metrics.

We create a dataset of 900 synthetic questions resolving in 2028 and create 3000 tuples in total from this dataset using the method described in Section 3.2, to evaluate the consistency of the forecasters in questions with a longer horizon, where it's not possible to have the ground truth resolutions. Examples of questions and the results for `gpt-4o` are in Appendix L.

We intend this dataset as a working prototype for a continual long-term forecasting benchmark.

## 5   ARBITRAGEFORECASTER: CAN WE DESIGN A MORE CONSISTENT FORECASTER?

Let $(x_1, \ldots x_n)$ be a question tuple for some consistency check $\mathcal{R}$, e.g. $(P, \neg P)$. Given forecasts $\mathbb{F}(x_1), \ldots \mathbb{F}(x_n)$, the arbitrage metric maximization problem in Equation 1 computes the following (as the argmax and max of the arbitrage respectively):

1. Improved forecasts $\mathbb{F}'(x_1), \ldots \mathbb{F}'(x_n)$ which are consistent, i.e. satisfy $\mathcal{S}$; and

2. The profit earned by an arbitrageur who bets these improved forecasts against the original ones – this is the actual metric.

This leads us to wonder: *can we use these "improved consistent forecasts" to build a new forecaster which builds on the base forecaster $\mathbb{F}$, but is more consistent on $\mathcal{R}$?*

We introduce: the `ArbitrageForecaster` with base $\mathbb{F}$ arbitraged on consistency check $\mathcal{R}$, denoted by $\langle \mathbb{F} \rangle_{\mathcal{R}}$, which computes its forecast on a question $x$ as follows:

1. Instantiates a tuple $(x_1, \ldots x_n)$ satisfying $\mathcal{R}$;

2. Queries $\mathbb{F}$ to obtain $\mathbb{F}(x_1), \ldots \mathbb{F}(x_n)$;

3. Arbitrages these base forecasts per Eq 1 and returns the arbitraged forecast for $x_1$.

Despite what one might assume, however, an `ArbitrageForecaster` is *not* "definitionally" consistent on the check it is arbitraged on, but rather its consistency must be investigated empirically. Suppose, for example, that a forecaster produces forecasts $\mathbb{F}(P) = 0.5$, $\mathbb{F}(\text{para}(P)) = 0.6$, $\mathbb{F}(\text{para}(\text{para}(P))) = 0.7$. Then $\mathbb{F}' := \langle \mathbb{F} \rangle_{\text{PARAPHRASE}}$ would produce forecasts $\mathbb{F}'(P) \approx 0.55$, $\mathbb{F}'(\text{para}(P)) \approx 0.65$, which are not consistent.

Appendix G contains a precise definition of `ArbitrageForecaster`, including the case of sequentially arbitraging on multiple checks $\langle \mathbb{F} \rangle_{[\mathcal{R}_1, \ldots \mathcal{R}_s]}$, and a theoretical discussion of its consistency

properties. In particular, we list strong theoretical reasons to expect consistency gains from *recursive* `ArbitrageForecaster` setups, i.e. $\langle\mathbb{F}\rangle_{\mathcal{R}}^r := \langle\langle\mathbb{F}\rangle_{\mathcal{R}}^{r-1}\rangle_{\mathcal{R}}$, in particular with NEGATION, as well as in a non-recursive `ArbitrageForecaster` with EXPEVIDENCE.

Due to these priorities and the high costs of running recursive `ArbitrageForecasters` (see Appendix G.1), we limited ourselves to studying only a small number of `ArbitrageForecaster` setups, with a limited number of checks rather than the whole list; specifically: $\langle g\rangle_N^r$, $\langle g\rangle_P^r$, $\langle g\rangle_{[N,P]}^r$, $\langle g\rangle_{[E]*s}$ where $g :=$ `gpt-4o-mini`, $N, P, E$ are NEGATION, PARAPHRASE, EXPEVIDENCE respectively, and $r$ and $s$ vary from 0 to 4.

The full results of our experiments with these forecasters are reported in Appendix G.2; our key takeaways from these preliminary runs look hopeful:

- In the case of the checks we tested, **arbitraging on a check indeed makes a forecaster more consistent on that check**, with increasing consistency gains with recursive depth, as shown in Fig 3. Crucially, this also applied when the arbitraging was on more than a single check: $\langle g\rangle_{[N,P]}^r$ did well on *both* NEGATION and PARAPHRASE; arbitraging on the next check did not increase inconsistency on the first. We are cautiously optimistic that this may extend to the full list of checks in Table 3.
- **This consistency gain was greatest with NEGATION, followed by PARAPHRASE, and lowest with EXPEVIDENCE.** This finding is in line with our hypothesis in Appendix G that `ArbitrageForecaster` would be particularly effective on consistency checks which are *symmetric*. and instantiate *deterministically*.
- **We do not observe reliable improvements on ground truth forecasting performance, or on consistency checks other than the ones we arbitrage on.** I.e. $\langle\mathbb{F}\rangle_{\mathcal{R}_1}$ does not reliably do better on $\mathcal{R}_2$.

# 6 FUTURE WORK

We have developed a comprehensive benchmark of *static consistency checks* for LLM forecasters, and demonstrated its correlation with ground truth accuracy, suggesting that our consistency metrics could serve as a proxy for accuracy when we do not have access to ground truth. We envision several directions in which our framework could be extended:

**Consistency in decision-making.** AI systems may be used not only to make forecasts that inform decisions, but also to take decisions directly. Here too, we can have a notion of inconsistency: for example, *intransitive preferences* [4] – and analogously, an inconsistent decision-maker may be exploited by an arbitrageur.

**Training for consistency.** Modulo consideration of the cost-benefit to safety, our methods could be used train LLMs for consistency, minimizing our violation metrics. This may or may not impact overall forecasting performance and other AI capabilities. One may also imagine an AlphaZero-style set-up, where an LLM $\mathbb{F}$ is trained on the outputs of $\langle\mathbb{F}\rangle^r$, i.e. a recursive `ArbitrageForecaster` wrapped around it.

**Further experiments with `ArbitrageForecaster`.** Most of our experiments with `ArbitrageForecaster` involved arbitraging on only a *single* check (apart from one experiment with both NEGATION and PARAPHRASE), due to the cost limitations described in G.1. It is easy to imagine how a bad forecaster could still overfit a single check: simply forecasting 50% probability for all questions will pass PARAPHRASE, EXPEVIDENCE and NEGATION – but we expect that being consistent under a variety of checks is difficult without a consistent world model. One approach to using more checks cheaply, particularly in training, may be to *randomly sample* a number of consistency checks for each question.

**Dynamic generation of consistency checks.** Although we found strong correlations between ground truth accuracy and consistency among existing LLM forecasters, our results with `ArbitrageForecaster` demonstrate that this isn't necessarily the case: it is possible to do well

---

[4]See e.g. Fishburn (1970) and the Von Neumann–Morgenstern utility theorem for an introduction to decision rationality.

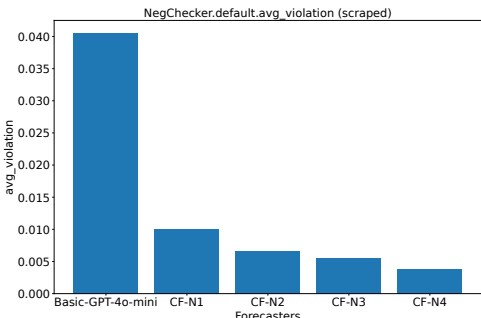

(a) Average violation of $\langle g \rangle_N^r$ (denoted CF-Nr) on NEGATION for $r$ from 0 to 4.

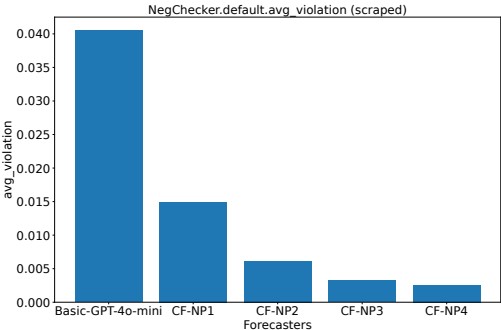

(b) Average violation of $\langle g \rangle_{NP}^r$ (denoted CF-NPr) on NEGATION for $r$ from 0 to 4.

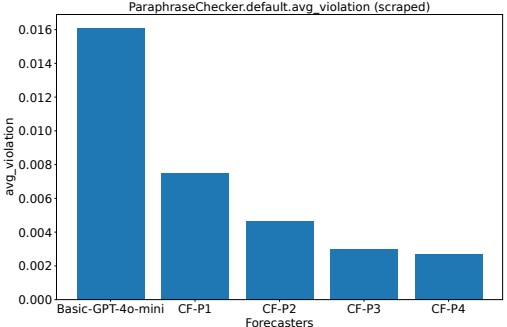

(c) Average violation of $\langle g \rangle_P^r$ (denoted CF-Pr) on PARAPHRASE for $r$ from 0 to 4.

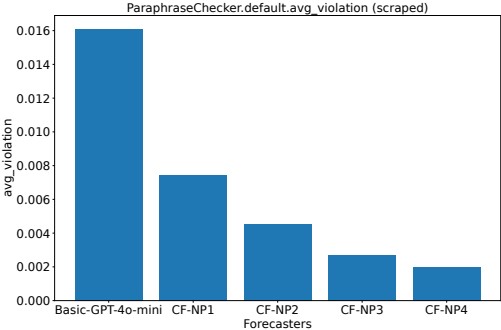

(d) Average violation of $\langle g \rangle_{NP}^r$ (denoted CF-NPr) on PARAPHRASE for $r$ from 0 to 4.

Figure 3: NEGATION and PARAPHRASE violations for various `ArbitrageForecaster` setups. In all captions, $g$ denotes `gpt-4o-mini`, $N, P$ denote NEGATION and PARAPHRASE respectively, and the definition of the `ArbitrageForecaster` setup is as in Def G.2.

on consistency without improving ground truth. In particular, this means that consistency as a training metric could be "Goodharted" by a learning AI model (Karwowski et al., 2023). One way to prevent this may be via adversarial training: i.e. have an adversarial agent instantiate consistency checks that it believes the agent will perform poorly on.

**Evaluating RAG-augmented forecasters.** We have conducted some preliminary experiments evaluating state-of-the-art forecasters such as Halawi et al. (2024). Unfortunately, we could not reproduce the system from Halawi et al. (2024) at the time of writing, due to deprecations in the Google News API (we could not obtain access to the alternative Newscatcher API). At the time of writing, we are not aware of other publicly-available LLM forecasting systems that are competitive with the results of Halawi et al. (2024) (there exist proprietary systems that may be competitive, such as FutureSearch (2024)). We thus leave the evaluation of better forecasters like Halawi et al. (2024) and Phan et al. (2024) to future work, once such forecasters are more widely available.

AUTHOR CONTRIBUTIONS

DP and APS developed consistency checks and the arbitrage and frequentist metrics. DP, AA, APS, and EW worked on the LLM question to evaluation pipeline. APS thought of and implemented `ArbitrageForecaster`. VB created the news-derived question dataset. AS and DP created the scraped question dataset. AA and DP created the 2028 synthetic question dataset. DP started and led the project. FT proposed correlating consistency with forecasting accuracy and advised the project. All authors helped with the writing. DP and APS wrote the first draft of the paper.

ACKNOWLEDGEMENTS

We thank Danny Halawi for extensive discussions and help with our setup. We thank Brendan Murphy, Ezra Karger, Fred Zhang, and Tatsunori Hashimoto for helpful discussions and feedback on the paper and forecasting in general. We thank Berkeley SPAR for connecting collaborators, and BERI for partially funding the project.

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

## A   RELATED WORK

**Metamorphic and consistency checks.** Checking logical properties of outputs of programs under semantic-preserving transforms has a long history (Chen et al., 1998). Before Fluri et al. (2023), variants of the consistency check framework were used for simple ML models (Christakis et al., 2022; Sharma and Wehrheim, 2020), vision (Hendrycks and Dietterich, 2019), and chat LLMs (Jang and Lukasiewicz, 2023), among other areas. Li et al. (2019) consider logical consistency checks beyond paraphrasing and negation for simple ML models.

**Forecasting and large language models.** LLMs and forecasting date back to Zou et al. (2022a) and Yan et al. (2023). Recently, strong performance of LLM forecasters on prediction market datasets has been claimed in (Halawi et al., 2024; Tetlock et al., 2024; Hsieh et al., 2024; Phan et al., 2024). Concurrent with our work, Karger et al. (2024) have introduced an automatically updating benchmark for forecasting.

**Scalable oversight and failures of superhuman AI.** The difficulty of evaluating models with superhuman performance in domains without a source of ground truth has long been acknowledged, and falls under the umbrella of *scalable oversight* (Amodei et al., 2016). Forecasting using AI oracles is one such domain. The use of consistency checks for scalable oversight has been studied in the simpler context of superhuman game AIs (Lan et al., 2022; Fluri et al., 2023), and in general question-answering tasks via debate (Irving et al., 2018).

**Consistency evaluations for LLMs.** Even on tasks where the ground truth is in principle knowable, consistency evaluations have long helped in cases where checking consistency is easier than getting the ground truth labels (Elazar et al., 2021; Li et al., 2023). Raj et al. (2023) measure paraphrasing consistency and ground truth accuracy on TruthfulQA (Lin et al., 2021) and find little to no correlation. Some forms of consistency checks have been applied on model internals to discover features related to LLM truthfulness and reliability (Burns et al., 2022; Kaarel et al., 2023).

## B    TABLE OF CONSISTENCY CHECKS

Table 3 includes all the consistency checks tested for in our benchmark. In most of them, we leave the logical relations between forecasting questions $\mathcal{R}$ implicit by constructing the sentences directly. For instance, $\mathcal{R}(x_1, x_2) := x_1 = \neg x_2$ is implied by simply writing $x_1, x_2$ as $P, \neg P$. In the rest of the appendix, we use the sentence-based ($P, Q$ instead of $x_1, x_2$) notation.

Table 3: Consistency checks and the logical consistency conditions.

| Name | Tuple | Condition ($\mathcal{S}$) |
|---|---|---|
| NEGATION | $(P, \neg P)$ | $\mathbb{F}(P) + \mathbb{F}(\neg P) = 1$ |
| PARAPHRASE $\mathcal{R}(P, Q) := P \iff Q$ | $(P, Q)$ | $\mathbb{F}(P) = \mathbb{F}(Q)$ |
| CONSEQUENCE $\mathcal{R}(P, Q) := P \implies Q$ | $(P, Q)$ | $\mathbb{F}(P) \leq \mathbb{F}(Q)$ |
| ANDOR | $(P, Q, P \wedge Q, P \vee Q)$ | $\mathbb{F}(P) + \mathbb{F}(Q) = \mathbb{F}(P \vee Q) + \mathbb{F}(P \wedge Q)$ |
| AND | $(P, Q, P \wedge Q)$ | $\max(\mathbb{F}(P) + \mathbb{F}(Q) - 1, 0) \leq \mathbb{F}(P \wedge Q) \leq \min(\mathbb{F}(P), \mathbb{F}(Q))$ |
| OR | $(P, Q, P \vee Q)$ | $\max(\mathbb{F}(P), \mathbb{F}(Q)) \leq \mathbb{F}(P \vee Q) \leq \min(1, \mathbb{F}(P) + \mathbb{F}(Q))$ |
| BUT | $(P, \neg P \wedge Q, P \vee Q)$ | $\mathbb{F}(P \vee Q) = \mathbb{F}(P) + \mathbb{F}(\neg P \wedge Q)$ |
| COND | $(P, Q\|P, P \wedge Q)$ | $\mathbb{F}(P)\mathbb{F}(Q\|P) = \mathbb{F}(P \wedge Q)$ |
| CONDCOND | $(P, Q\|P, R\|(P \wedge Q), P \wedge Q \wedge R)$ | $\mathbb{F}(P)\mathbb{F}(Q\|P)\mathbb{F}(R\|P \wedge Q) = \mathbb{F}(P \wedge Q \wedge R)$ |
| EXPEVIDENCE | $(P, Q, P\|Q, P\|\neg Q)$ | $\mathbb{F}(P) = \mathbb{F}(P\|Q)\mathbb{F}(Q) + \mathbb{F}(P\|\neg Q)(1 - \mathbb{F}(Q))$ |

The consistency checks in Table 3 represent core logical relationships between probabilities, but many other forms of consistency checks are possible. Here are two examples that could extend our framework:

- **Comparative checks:** Building on generator-validator checks from Li et al. (2023), we could ask a forecaster to predict both $\mathbb{F}(P), \mathbb{F}(Q)$, and separately whether $P$ or $Q$ is more likely. The forecaster's probability estimates should match their comparative judgment.
- **Monotonicity checks:** Fluri et al. (2023) propose a variant of CONSEQUENCE for real-valued quantities, where predictions must respect the monotonic ordering of a sequence of future values. This connects to *scope insensitivity* (Kahneman et al., 2000), a cognitive bias where humans fail to scale probability estimates appropriately with the magnitude of outcomes.

We do not include a specific consistency check for Bayesian updates, as conditional probabilities are already covered by COND, CONDCOND, and EXPEVIDENCE.

# C  DATA TYPES USED IN OUR PIPELINE

## C.1  FORECASTING QUESTIONS

Figure 4 shows the data stored on forecasting questions. Of these, only *title* and *body* are shown to the forecaster.

> **Forecasting question Data Type**
>
> - **id**: Universally Unique Question Identifier (UUID), auto-generated using a default factory.
> - **title**: Title of the forecasting question.
> - **body**: Detailed resolution criteria, background information, etc.
> - **resolution_date**: The date when the question is expected to be resolved. We only consider questions that have a clear date when the resolution should be decided.
> - **question_type**: Type of the forecasting question; in this paper, only *binary* and *conditional-binary*. Options not used in this paper include *multiple-choice*, *interval*, *continuous-value*, or *opinion*.
> - **data_source**: Source of the question, either the website from which it was scraped or *synthetic*.
> - **created_date**: The date when the question was created, or *null* if not important for the meaning of the question.
> - **url**: URL of the source if the question was scraped, else *null*.
> - **metadata**: Any additional information, e.g., *topics*, *tags*, *category*; but also data fields specific to Metaculus, Manifold, etc; the source articles for NewsAPI-generated questions; or instantiation metadata for questions in consistency tuples.
> - **resolution**: A boolean indicating whether the question resolves to YES or NO, or *null* if unresolved.

Figure 4: Description of the forecasting question data type.

For instance, a forecasting question from Metaculus, such as the one shown in Figure 5, will be stored in the form depicted in Figure 6 using our method. The original question, which asks whether SpaceX will land people on Mars before 2030, is presented with detailed conditions for resolution, including specific criteria such as the confirmation of the landing by SpaceX and the completion of an extravehicular activity (EVA) on the Martian surface.

The data type in Figure 4 is compatible (after appropriate processing) with scraped questions from Metaculus and Manifold, and standardization helps with synthetic question generation and tuple instantiation. We do not include information about human forecasts because we explicitly focus on evaluation without relying on any human-generated probabilities.

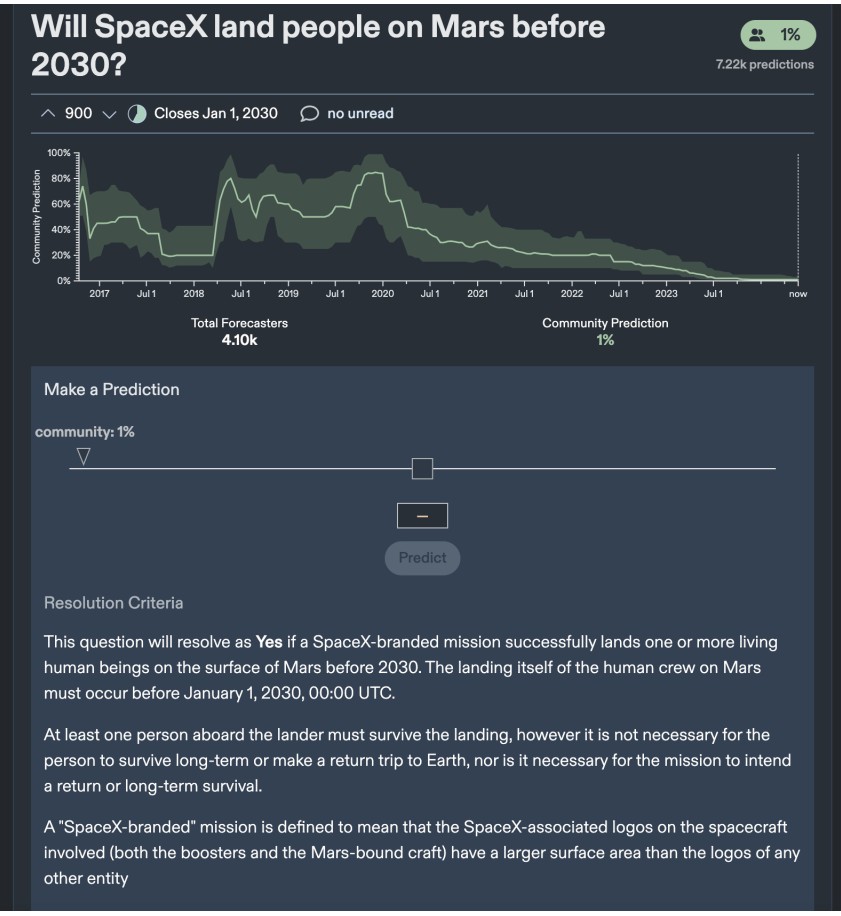

Figure 5: Example of a question on the Metaculus platform.

---

**Example forecasting question (scraped)**

- **id**: 07b11b15-6872-4280-a94f-17b6d15a1b8a
- **title**: Will SpaceX land people on Mars before 2030?
- **body**: This question will resolve as Yes if SpaceX successfully lands at least one human on the surface of Mars on or before December 31, 2030. The landing must be confirmed by SpaceX through an official announcement or live broadcast. The human(s) must be alive upon landing and must perform at least one extravehicular activity (EVA) on the Martian surface, which must be documented and released to the public. In the event of a dispute regarding the success of the mission, the resolution will defer to the judgment of an international space agency such as NASA or ESA. If no landing attempt is made by the specified date, or if all attempts fail to meet the above criteria, the question will resolve as No.
- **resolution_date**: 2030-12-31 23:59:59+00:00
- **question_type**: binary
- **data_source**: metaculus
- **url**: https://www.metaculus.com/questions/349
- **metadata**:
  - **topics**:
    * **id**: 184, **slug**: elon-musk, **name**: Elon Musk, **link_id**: 27681, **num_questions**: 159
    * **id**: 485, **slug**: spacex-reusable-launch-system-development-program, **name**: SpaceX reusable launch system, **link_id**: 27682, **num_questions**: 130
    * **id**: 1365, **slug**: spacex, **name**: SpaceX, **link_id**: 75197, **num_questions**: 112
    * **id**: 564, **slug**: colonization-of-mars, **name**: Colonization of Mars, **link_id**: 27683, **num_questions**: 70
    * **id**: 1768, **slug**: spacex-mars-transportation-infrastructure, **name**: SpaceX Mars transportation infrastructure, **link_id**: 40982, **num_questions**: 5
- **resolution**: null

Figure 6: Example of a forecasting question scraped from Metaculus.

By processing this question through our pipeline, we retain all relevant details, such as the resolution date and specific criteria for a binary outcome, while structuring the data in a more standardized format to facilitate further analysis. Additionally, associated metadata, including related topics and links to other questions, is also preserved.

---

**Example forecasting question (synthetic)**

- **id**: 4b98368c-6287-47e0-8f9e-5917e2a24a3d
- **title**: Will Russia launch a manned mission to the Moon before 2030?
- **body**: This question will resolve as Yes if, before January 1, 2030, the Russian Federation successfully launches and completes a manned mission to the Moon, where 'successful' is defined as a mission where astronauts land on the lunar surface and return safely to Earth. The mission must be officially recognized by Roscosmos or another authoritative space agency. In the event of a joint mission involving Russia and other countries, the mission will still resolve as Yes if Russian astronauts are part of the crew that lands on the Moon. If no such mission is launched, or if a mission is launched but does not meet the above criteria, the question will resolve as No. In the case of ambiguity or lack of clear public information by the resolution date, the question will resolve as No unless official statements or evidence are provided by Roscosmos or an equivalent authoritative body that confirm the mission's success as per the defined criteria.
- **resolution_date**: 2030-12-31 23:59:59+00:00
- **question_type**: binary
- **data_source**: synthetic
- **url**: null
- **metadata**:
  - **tags**:
    * Russia
  - **categories**:
    * Space
- **resolution**: null

---

Figure 7: Example of a synthetic forecasting question. All question generations are seeded with the *metadata* field.

As an example, we also show a forecasting question generated synthetically using the source tags "Russia" and "Moon" could ask whether Russia will launch a manned mission to the Moon by 2030. The structure and format of this synthetic question, as illustrated in Figure 7, mirror those of real forecasting questions while maintaining the essential metadata for context.

## C.2 EXAMPLES OF INSTANTIATED TUPLES

In the following examples, we focus on the question title for clarity. Figure 8 illustrates an instantiated AND tuple, starting from forecasting questions (**P** and **Q**) that address distinct events regarding artificial intelligence policy in the U.S. and Canada, together with a conjunction question (**P_and_Q**) about their joint occurrence by a specified date. Figure 9 presents an instantiated EXPEVIDENCE tuple, examining the global space industry's revenue potential alongside the political dynamics in the U.S. House of Representatives, including conditional questions that evaluate the influence of one event on another.

We note that making the detailed resolution criteria ("body" field) actually correspond to the composite event is not straighforward, and is only in reach of the newest generations of LLMs. A different design option would be to just list the original questions and resolution criteria separately in the "body" field, and then say what the logical operation is. We opt against it for two reasons:

- A separate, unnatural format for composite questions might induce qualitatively different behaviors in LLM forecasters.
- Future works in this framework might not rely on simple logical operations, but rather on an advanced LLM grader that computes "do these forecasts make sense taken together". Our current design allows for an easier extension to this direction.

---

**Example tuple (AND)**

- **P**:
    - **title**: Will the United States pass a federal law regulating the ethical use of artificial intelligence in energy management before January 1, 2028?
- **Q**:
    - **title**: Will Canada implement a nationwide artificial intelligence policy before January 1, 2028?
- **P_and_Q**:
    - **title**: Will both of the following occur before January 1, 2028: (a) the United States passes a federal law regulating the ethical use of artificial intelligence in energy management and (b) Canada implements a nationwide artificial intelligence policy?

---

Figure 8: Example of an instantiated AND forecasting question tuple. We omit the rest of the fields for brevity.

---

**Example tuple (EXPEVIDENCE)**

- **P**:
    - **title**: Will the global space industry generate annual revenues exceeding $1 trillion by the end of 2027?
- **Q**:
    - **title**: Will the Democratic Party gain a majority in the US House of Representatives after the 2026 midterm elections?
- **P_given_Q**:
    - **title**: Given the Democratic Party gains a majority in the US House of Representatives after the 2026 midterm elections, will the global space industry generate annual revenues exceeding $1 trillion by the end of 2027?
- **P_given_not_Q**:
    - **title**: Conditional on the Democratic Party failing to gain a majority in the US House of Representatives after the 2026 midterm elections, will the global space industry generate annual revenues exceeding $1 trillion by the end of 2027?

---

Figure 9: Example of an instantiated EXPEVIDENCE forecasting question tuple. We omit the rest of the fields for brevity.

# D ARBITRAGE AS A VIOLATION METRIC

For the following definition we use a slightly more general notation than in the main body, to convey that our methods could be generalized beyond binary forecasting questions.

*Notation.* Let $\mathrm{Prop}$ denote the set of forecasting questions we are interested in, $\Theta$ denote the set of possible outcomes/resolutions for an individual question, and $\Delta\Theta$ denote the set of probability distributions on $\Theta$. A *Forecaster* is a map $\mathbb{F} : \mathrm{Prop} \to \Delta\Theta$. For conditional questions that can resolve to $\mathrm{None}$, we also have optional resolutions $\Theta' := \Theta \cup \{\mathrm{None}\} = \{\top, \bot, \mathrm{None}\}$.

The arbitrage metric may be seen as being motivated by Dutch Book Arguments for probabilistic consistency rules (see e.g. Vineberg (2022)). Imagine the forecaster's predictions $\mathbb{F}(x_1), \ldots \mathbb{F}(x_n)$ were prices offered by a bookie on prediction markets for sentences $x_1, \ldots x_n$. If these probabilities are inconsistent, then there are bets that an arbitrageur can make that guarantee a profit in *all possible (consistent) worlds regardless of the individual outcomes*. For example, if $x_1, x_2$ are two sentences such that $x_1 \iff x_2$, but the bookie prices $\mathbb{F}(x_1) < \mathbb{F}(x_2)$, then an arbitrageur can simply buy $x_1$ and sell $x_2$ to make a risk-free profit.

However, if the bookie never changes their prices in response to trades, the arbitrageur can make an infinite amount of profit with its strategy. This is neither realistic nor useful for creating a metric to measure inconsistency. Instead, we turn to *market scoring rules*, introduced in Hanson (2002)), where the bookie is a *market-maker* who updates market prices in a way that ensures that the reward for moving the market price of a sentence that resolves True from $p_0$ to $p'$ is given by a *proper scoring rule* [5] $s(p') - s(p_0)$. We then define our inconsistency metric to be the minimum profit an arbitrageur can guarantee against such a market-maker, if the latter offers inconsistent probabilities $\mathbb{F}(x_1), \ldots \mathbb{F}(x_n)$.

**Definition D.1** (Arbitrage-based Violation Metric). Let $\mathcal{R} : \mathrm{Prop}^n \to \{\top, \bot\}$ be an n-ary relation such that $\mathcal{R}(\theta(x_1), \ldots \theta(x_n))$ is satisfied by the ground-truth resolutions $\theta : \mathrm{Prop} \to \Theta$ for all tuples $(x_1, \ldots x_n)$. [6] Let $s : \mathrm{Prop} \times \Theta \times [0,1] \to \mathbb{R}$ be a proper scoring rule that gives the score earned based on the probability assigned to the true resolution, e.g. $s(x, \theta, p(\theta)) = \log p(\theta)$. Let $(x_1, \ldots x_n) \in \mathrm{Prop}^n$ be a question tuple, and denote $\Omega := \{\omega \in \Theta'^n \mid \mathcal{R}(\omega)\}$ the set of possible consistent resolutions (including $\mathrm{None}$ resolutions) of this tuple. Then for forecasts $(\mathbb{F}(x_1), \ldots \mathbb{F}(x_n))$ the arbitraged forecasts $\mathcal{A}(\mathbb{F}(x_1), \ldots \mathbb{F}(x_n)) = (p_1 \ldots p_n)$ and the minimum guaranteed profit of the arbitrageur $\mathcal{V}(\mathbb{F}(x_1), \ldots \mathbb{F}(x_n))$ are given by:

$$(\arg\max, \max)_{p \in \Delta\Theta^n} \min_{\omega \in \Omega} \sum_{i=1}^{n} s(x_i, \omega_i, p_i(\omega_i)) - s(x_i, \omega_i, \mathbb{F}(x_i)(\omega_i)) \quad (2)$$

Where by convention, any score on a resolution $\omega_i = \mathrm{None}$ is taken to be 0.

Definition D.1 is presented in full generality: $p$ and $\mathbb{F}(x_i)$ here are *probability distributions* on $\Theta$. Breaking it down: each $s(x_i, \omega_i, p_i(\omega_i)) - s(x_i, \omega_i, \mathbb{F}(x_i)(\omega_i))$ gives the arbitrageur's profit on the market for question $x_i$, given that it resolves $\omega_i$. The profit is summed across all markets in the tuple, and then minimized over all consistent worlds; this minimum is maximized across all possible arbitrageur bets.

It is helpful to explicitly state Eq 2 in the case of binary forecasting questions, as follows.

$$(\arg\max, \max)_{p \in [0,1]^n} \min_{\omega \in \Omega} \sum_{i=1}^{n} (s(p_i) - s(\mathbb{F}(x_i))) \delta_{\omega(i)=\top} + (s(1-p_i) - s(1-\mathbb{F}(x_i))) \delta_{\omega(i)=\bot} \quad (3)$$

We will illustrate our violation metric with three specific examples, for PARAPHRASE, NEGATION and COND. For other consistency checks, the math becomes too convoluted and we use a numerical method in our project code.

---

[5] A proper scoring rule (Savage, 1971), is one that incentivizes honest reporting of probabilities: widely used proper scoring rules include the Brier score $(1 - p)^2$ and the logarithmic scoring rule $-\log p$.

[6] This is well-defined because resolutions can be taken as a subset $\Theta \subseteq \mathrm{Prop}$, by treating them as forecasting questions that always resolve to themselves by definition. For example, the forecasting question $\top$ is always worth \$1 and the forecasting question $\bot$ is always worth \$0.

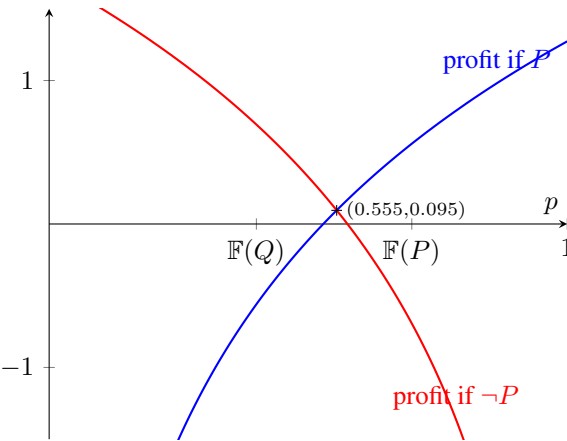

Figure 10: Profit earned by the arbitrageur in case of inconsistency over ParaphraseChecker, taking $s(p) = \log(p)$ and $\mathbb{F}(P), \mathbb{F}(Q) = 0.7, 0.4$ in (4).

.

## D.1 PARAPHRASECHECKER

Let $P$ and $Q$ be equivalent sentences, and suppose that the forecaster produces forecasts $\mathbb{F}(P)$ and $\mathbb{F}(Q)$. A trader who instead brings prices to $\mathbb{F}'(P) = \mathbb{F}'(Q) = p$ for both questions earns a combined profit on both questions:

$$\begin{cases} s\,(p) - s\,(\mathbb{F}(P)) + s\,(p) - s\,(\mathbb{F}(Q)) & \text{if } P \\ s\,(1-p) - s\,(1-\mathbb{F}(P)) + s\,(1-p) - s\,(1-\mathbb{F}(Q)) & \text{if } \neg P \end{cases} \tag{4}$$

For this first example, we can graph this profit as a function of $p$ for illustration, shown in Fig. 10 – demonstrating that any $p \in (0.529, 0.576)$ is profitable for the arbitrageur, and further that the arbitrageur can *guarantee* a minimum profit of $0.095$ regardless of the outcome of $P$ by choosing the consistent probability $p = 0.555$.

We may compute this intersection analytically:

$$s\,(p) - s\,(\mathbb{F}(P)) + s\,(p) - s\,(\mathbb{F}(Q)) = s\,(1-p) - s\,(1-\mathbb{F}(P)) + s\,(1-p) - s\,(1-\mathbb{F}(Q))$$

$$2\log\frac{p}{1-p} = \log\frac{\mathbb{F}(P)\mathbb{F}(Q)}{(1-\mathbb{F}(P))(1-\mathbb{F}(Q))}$$

$$p = \frac{\sqrt{\mathbb{F}(P)\mathbb{F}(Q)}}{\sqrt{\mathbb{F}(P)\mathbb{F}(Q)} + \sqrt{(1-\mathbb{F}(P))(1-\mathbb{F}(Q))}}$$

Substituting this back into either expression in (4) we get the expression for the arbitrage:

$$\mathcal{V}(\mathbb{F}(P), \mathbb{F}(Q)) = -2\log\left(\sqrt{\mathbb{F}(P)\mathbb{F}(Q)} + \sqrt{(1-\mathbb{F}(P))(1-\mathbb{F}(Q))}\right) \tag{5}$$

As a bonus, this can straightforwardly be extended to the multi-question paraphrasing check: $(P_1 \iff \cdots \iff P_n) \implies (\mathbb{F}(P_1) = \cdots = \mathbb{F}(P_n))$. Here the corresponding possible profits are:

$$\begin{cases} ns\,(p) - \sum s\,(\mathbb{F}(P_i)) & \text{if } P \\ ns\,(1-p) - \sum s\,(1-\mathbb{F}(P_i)) & \text{if } \neg P \end{cases} \tag{6}$$

Equating them and solving for $p$, we get:

$$\log \frac{p}{1-p} = \frac{1}{n} \sum_i \log \frac{\mathbb{F}(P_i)}{1 - \mathbb{F}(P_i)} \tag{7}$$

$$p = \frac{\Delta}{\Delta + 1} \text{ where } \Delta = \left[ \prod_i \frac{\mathbb{F}(P_i)}{1 - \mathbb{F}(P_i)} \right]^{1/n} \tag{8}$$

Observe that the arbitraged probability is simply the arithmetic mean in log-odds space! One may wonder if the violaton is some kind of variance measure in log-odds space, but this does not seem to be the case:

$$\mathcal{V}(\mathbb{F}(P_1), \dots \mathbb{F}(P_n)) = -n \log \left[ \left( \prod \mathbb{F}(P_i) \right)^{1/n} + \left( \prod (1 - \mathbb{F}(P_i)) \right)^{1/n} \right] \tag{9}$$

## D.2 NegChecker

Suppose the forecaster produces forecasts $\mathbb{F}(P)$ and $\mathbb{F}(\neg P)$. A trader who instead brings prices to $\mathbb{F}'(P) = p$, $\mathbb{F}'(\neg P) = 1 - p$ earns a combined profit on both questions:

$$\begin{cases} s(p) - s(\mathbb{F}(P)) + s(p) - s(1 - \mathbb{F}(\neg P)) & \text{if } P \\ s(1-p) - s(1 - \mathbb{F}(P)) + s(1-p) - s(\mathbb{F}(\neg P)) & \text{if } \neg P \end{cases} \tag{10}$$

Equating them and solving as before,

$$2 \log \frac{p}{1-p} = \log \frac{\mathbb{F}(P)(1 - \mathbb{F}(\neg P))}{(1 - \mathbb{F}(P))\mathbb{F}(\neg P)}$$

$$p = \frac{\sqrt{\mathbb{F}(P)(1 - \mathbb{F}(\neg P))}}{\sqrt{\mathbb{F}(P)(1 - \mathbb{F}(\neg P))} + \sqrt{(1 - \mathbb{F}(P))\mathbb{F}(\neg P)}}$$

Substituting into (10), we get:

$$\mathcal{V}(\mathbb{F}(P), \mathbb{F}(\neg P)) = -2 \log \left( \sqrt{\mathbb{F}(P)(1 - \mathbb{F}(\neg P))} + \sqrt{(1 - \mathbb{F}(P))\mathbb{F}(\neg P)} \right) \tag{11}$$

The similarity of these results to PARAPHRASE is suggestive: both the arbitraged probability and the violation for NEGATION can be derived from PARAPHRASE simply replacing $\mathbb{F}(Q)$ with $1 - \mathbb{F}(\neg P)$, seeing the latter as the "probability implied for $P$ by $\neg P$". This raises the natural question: Can *all* consistency checks be reduced to the case of PARAPHRASE arbitraging $\mathbb{F}(P)$ against an "implied probability" for $P$?

However, as we will see, COND shows that this approach does not always hold. Its violation expression depends on more than just $\mathbb{F}(P)$ and $\mathbb{F}(P \wedge Q)/\mathbb{F}(Q \mid P)$, so there is no single, neat interpretation akin to "arithmetic mean in the log-odds space."

## D.3 CondChecker

Suppose the forecaster produces forecasts $\mathbb{F}(P)$, $\mathbb{F}(Q \mid P)$, $\mathbb{F}(P \wedge Q)$. The possible outcomes $\Omega$ are $(P, Q \mid P, P \wedge Q) \mapsto (\top, \top, \top), (\top, \bot, \bot), (\bot, \text{None}, \bot)$. Consider an arbitrageur who makes bets $\mathbb{F}'(P) = p$, $\mathbb{F}'(Q \mid P) = q$, $\mathbb{F}'(P \wedge Q) = pq$.

In each outcome:

$$\begin{cases} s\left(p\right) - s\left(\mathbb{F}(P)\right) + s\left(q\right) - s\left(\mathbb{F}(Q\mid P)\right) + s\left(pq\right) - s\left(\mathbb{F}(P\wedge Q)\right) & \text{if } P, Q \\ s\left(p\right) - s\left(\mathbb{F}(P)\right) + s\left(1-q\right) - s\left(1-\mathbb{F}(Q\mid P)\right) + s\left(1-pq\right) - s\left(1-\mathbb{F}(P\wedge Q)\right) & \text{if } P, \neg Q \\ s\left(1-p\right) - s\left(1-\mathbb{F}(P)\right) + s\left(1-pq\right) - s\left(1-\mathbb{F}(P\wedge Q)\right) & \text{if } \neg P \end{cases}$$

$$(12)$$

Equating these and rearranging:

$$\begin{cases} \frac{1-p}{p(1-q)} = \frac{1-\mathbb{F}(P)}{\mathbb{F}(P)(1-\mathbb{F}(Q|P))} =: A \\ \frac{1-q}{q}\frac{1-pq}{pq} = \frac{(1-\mathbb{F}(Q|P))(1-\mathbb{F}(P\wedge Q))}{\mathbb{F}(Q|P)\mathbb{F}(P\wedge Q)} =: B \end{cases}$$

Solving, where we indicate the right-hand-sides of each equation above by $A$ and $B$ respectively:

$$p = \frac{1 + \sqrt{B/(A+1)}}{1 + \sqrt{B\cdot(A+1)}}$$

$$q = \frac{1}{1 + \sqrt{B/(A+1)}}$$

$$pq = \frac{1}{1 + \sqrt{B\cdot(A+1)}}$$

Substituting back into 12 and simplifying:

$$\mathcal{V}(\mathbb{F}(P), \mathbb{F}(Q\mid P), \mathbb{F}(P\wedge Q))$$
$$= -2\log\left(\sqrt{\mathbb{F}(P)\mathbb{F}(Q\mid P)\mathbb{F}(P\wedge Q)} + \sqrt{(1-\mathbb{F}(P)\mathbb{F}(Q\mid P))(1-\mathbb{F}(P\wedge Q))}\right).$$

### D.4 NUMERICAL ESTIMATION

Explicitly deriving the violation metrics for other checkers from Equation (2) is infeasible by hand, and the expressions yielded by SymPy are very convoluted. For these checks, we use a numerical algorithm based on solving a differential equation for $p_i(t)$, as detailed below.

The arbitraging process may be understood as adjusting market prices in such a way that the scores in each possible $\omega \in \Omega$ remain equal throughout the process – i.e. such that their *derivatives* remain equal. For derivatives $p_i'(t)$ of the prices, the derivatives of each score $s_\omega'(t)$ are:

$$s_\omega'(t) = \begin{bmatrix} a_{\omega 1}(p_1) & \cdots & a_{\omega n}(p_n) \end{bmatrix} \cdot \begin{bmatrix} p_1'(t) \\ \vdots \\ p_n'(t) \end{bmatrix}$$

where

$$a_{\omega i}(p_i) = \begin{cases} s'(p_i) & \text{if } \omega_i = \top, \\ -s'(1-p_i) & \text{if } \omega_i = \bot, \\ 0 & \text{if } \omega_i = \text{N/A} \end{cases}$$

Then, where $A(\mathbf{p}) = [a_{\omega i}(p_i)]$ (with $\Omega$ rows and $n$ columns), we have $\mathbf{s}'(t) = A(\mathbf{p})\mathbf{p}'(t)$. We want $\mathbf{s}'(t)$ to be a multiple of $\begin{bmatrix} 1 & \cdots & 1 \end{bmatrix}$ to ensure it is the same in all outcomes $\omega$. Since the coefficient of proportionality only controls how quickly the process converges, we can solve $\mathbf{p}'(t) = A^{-1}\mathbf{s}'(t)$. The dynamics are then:

$$p_i(0) = \mathbb{F}(x_i) \quad \text{(initial conditions)}$$

$$\mathbf{p}'(t) = A(\mathbf{p})^{-1} \begin{bmatrix} 1 \vdots 1 \end{bmatrix}$$

Consistency is reached when $\det A$ reaches 0.

## E   FREQUENTIST CONSISTENCY METRIC

In a deterministic world, we cannot let any inconsistency pass; every time we prove any rule of probability does not hold exactly, we must discard the forecaster as flawed. This is too strict for the consistency check framework to be useful. Instead, we propose a violation metric and the corresponding inconsistency threshold based on statistical hypothesis testing.

Assume that each event $P$ has a true probability value $\mathbb{T}(P)$, say under some world model that accounts for aleatoric uncertainty.

**Definition E.1** (Frequentist consistency). A frequentist-consistent forecaster $\mathbb{F}$ samples a Gaussian estimate $\mathbb{T}(P) + \varepsilon$ of each event $P$, with variance $\sigma^2 \mathbb{T}(P)(1 - \mathbb{T}(P))$ for a hyperparameter $\sigma^2$:

$$\mathbb{F}(P) - \mathbb{T}(P) \sim \mathsf{N}\left(0, \sigma^2 \mathbb{T}(P)(1 - \mathbb{T}(P))\right) \quad \text{independently for all events } P. \tag{13}$$

This is principled from the frequentist perspective. Consider a forecaster that just samples the (relevant subset of) the world $n$ times using the best available world simulator, and estimates the probability of each event $P$ as the proportion of times that $P$ occurs in the $n$ samples. If we estimate the probability as the average chance of an event $P$ with true probability $p$ occurring out of $n$ times, then this estimate has a scaled binomial distribution with mean $p$ and variance $p(1 - p)/n$. To reach Equation (13), replace the averaged binomial with the Gaussian of the same variance, and denote $\sigma^2 := 1/n$.

This simple model enables us to derive hypothesis tests for each of the consistency checks described in Table 3. The null hypothesis is always that the forecaster is frequentist-consistent. Note that $\sigma^2$ is not our estimate of the variance of any forecaster; it is just a hyperparameter that controls how strict our null hypothesis is. We leave estimating the variance of a particular forecaster and testing frequentist consistency based on that alone to future work.

**Notation**   The expression $a\mathsf{N}(0, c^2)$ denotes a Gaussian random variable with mean 0 and variance $a^2 c^2$. The expression $a\mathsf{N}(0, c^2) + b\mathsf{N}(0, c^2)$ denotes a Gaussian random variable with mean 0 and variance $a^2 c^2 + b^2 c^2$. All sums range over the cyclic permutations of the variables under the sum. All $\mathsf{N}(0, c^2)$ terms appearing with the same power of $\sigma$ are independent. Two $\mathsf{N}(0, c^2)$ terms appearing with a different power of $\sigma$ may be correlated; this is not important for our purposes, since we discard high-order powers of $\sigma$.

**Bootstrapping the true probability**   The final expressions for hypothesis test statistics might involve the true probability $\mathbb{T}(P)$. It is not available, so we just plug in $\mathbb{F}(P)$ for $\mathbb{T}(P)$ in the end. If we had a prior on $\mathbb{T}(P)$, we could combine it with $\mathbb{F}(P)$ to get a more robust estimate.

**NEGATION**   We take the violation metric and the corresponding threshold as to produce a hypothesis test against this:

$$\mathbb{F}(P) + \mathbb{F}(\neg P) - 1 = \mathbb{T}(P) + \varepsilon_1 + \mathbb{T}(\neg P) + \varepsilon_2 - 1 = \varepsilon_1 + \varepsilon_2$$
$$\sim \mathsf{N}\left(0, \sigma^2(\mathbb{T}(P)(1 - \mathbb{T}(P)) + \mathbb{T}(\neg P)(1 - \mathbb{T}(\neg P)))\right)$$

We estimate the unknown $\mathbb{T}$ values with the corresponding $\mathbb{F}$ estimates. Note that, although $\mathbb{T}(P) = 1 - \mathbb{T}(\neg P)$, it is of course not necessarily the case that $\mathbb{F}(P) = 1 - \mathbb{F}(\neg P)$.

The error distribution is $\sigma \mathsf{N}\left(\mathbb{F}(P)(1 - \mathbb{F}(P)) + \mathbb{F}(\neg P)(1 - \mathbb{F}(\neg P))\right)$, and the two-sided test is

$$|\mathbb{F}(P) + \mathbb{F}(\neg P) - 1| < \gamma \sigma \sqrt{(1 - \mathbb{F}(P))\mathbb{F}(P) + (1 - \mathbb{F}(\neg P))\mathbb{F}(\neg P)}$$

for some scale factor $\gamma$ (number of standard deviations) that scales the power of the test. For example, $\gamma = 2.58$ gives a 99%-confidence interval.

We now want to compute some *consistency violation metric* that makes inconsistency comparable across different checks. The natural idea is to aggregate all terms dependent on $\mathbb{F}$ to one side; and make the hypothesis test be just some threshold on the computed violation metric.

It is possible that the denominator of the resulting expression is 0 when the forecaster is certain and $\mathbb{F}$ is 0 or 1; to avoid division with zero, we add a small regularization term $\beta_{\mathrm{MIN}} = 10^{-3}$. See the last paragraph of this section for a discussion of hyperparameters.

Our consistency violation metric is then:

$$v_{\text{NEGATION}} = \frac{|\mathbb{F}(P) + \mathbb{F}(\neg P) - 1|}{\sqrt{(1 - \mathbb{F}(P))\mathbb{F}(P) + (1 - \mathbb{F}(\neg P))\mathbb{F}(\neg P) + \beta_{\text{MIN}}}}.$$

The hyperparameter $\sigma^2$ determines how strict we are with rejecting inconsistencies which could be attributed to "noisy" predictions. Note that the violation metric itself does not depend on $\sigma^2$.

A violation (inconsistency), therefore, occurs when:

$$v_{\text{NEGATION}} > \gamma\sigma.$$

**CONDCOND** This is a more complex consistency check; we derive the hypothesis test and violation metric in detail below. For the other checks, we just report the short derivation.

$$(a, b, c, d) = (\mathbb{T}(P), \mathbb{T}(Q \mid P), \mathbb{T}(R \mid P \wedge Q), \mathbb{T}(P \wedge Q \wedge R))$$
$$(a', b', c', d') = (\mathbb{F}(P), \mathbb{F}(Q \mid P), \mathbb{F}(R \mid P \wedge Q), \mathbb{F}(P \wedge Q \wedge R))$$

We can write:

$$\mathbb{F}(P) = \mathsf{N}\left(0, \sigma^2 a(1 - a)\right) + a,$$
$$\mathbb{F}(Q \mid P) = \mathsf{N}\left(0, \sigma^2 b(1 - b)\right) + b,$$
$$\mathbb{F}(R \mid P \wedge Q) = \mathsf{N}\left(0, \sigma^2 c(1 - c)\right) + c,$$
$$\mathbb{F}(P \wedge Q \wedge R) = \mathsf{N}\left(0, \sigma^2 d(1 - d)\right) + d$$

We now compute the difference of the two expressions that should be equal. All sums and products are cyclic over $a$, $b$, $c$.

$$\mathbb{F}(P)\mathbb{F}(Q \mid P)\mathbb{F}(R \mid P \wedge Q) - \mathbb{F}(P \wedge Q \wedge R) = abc - d$$
$$+ \sigma\left(\sum_a bc\mathsf{N}(0, a(1 - a)) - \mathsf{N}(0, d(1 - d))\right)$$
$$+ \sigma^2 \sum_a \mathsf{N}(0, b(1 - b))\mathsf{N}(0, c(1 - c))$$
$$+ \sigma^3 \prod_a \mathsf{N}(0, a(1 - a)).$$

In the above, all Gaussians with the same variance are identical, and all other combinations are independent. As $abc - d = 0$ by the law of total probability, the leading error term is next to $\sigma$. This is a Gaussian with mean 0 and standard deviation:

$$\sigma\sqrt{\sum_a b^2 c^2 a(1 - a) + d(1 - d)} = \sigma\sqrt{abc \sum_a bc(1 - a) + d(1 - d)}$$

We now discard the terms of $\sigma^2$, $\sigma^3$, and in general any higher order power of $\sigma$. This is principled because the coefficients can always be (in some confidence interval) upper bounded by a constant independent of $\sigma$. Hence, if $\sigma$ is small enough, the resulting test will be very close to the true hypothesis test.

We do not have the true probabilities $a$, $b$, $c$, $d$, so we just plug in $(a', b', c', d') = (\mathbb{F}(P), \mathbb{F}(Q \mid P), \mathbb{F}(R \mid P \wedge Q), \mathbb{F}(P \wedge Q \wedge R))$. [7] Thus the hypothesis test is (where the sum is cyclic over $a'$, $b'$, $c'$):

---

[7] Depending on how we use the relation $abc = d$, we can end up with different expressions in the end. We choose the one that, after plugging in, (i) yields an expression for variance that is always nonnegative, and (ii) is not a polynomial multiple of any single value of $\mathbb{F}$.

$$|a'b'c' - d'| > \gamma\sigma\sqrt{a'b'c'\sum_{a'} b'c'(1-a') + d'(1-d')}$$

Our violation metric is then:

$$v_{\text{CONDCOND}} = \frac{|a'b'c' - d'|}{\sqrt{a'b'c'\sum_{a'} b'c'(1-a') + d'(1-d') + \beta_{\text{MIN}}}}.$$

where again $(a', b', c', d') = (\mathbb{F}(P), \mathbb{F}(Q \mid P), \mathbb{F}(R \mid P \wedge Q), \mathbb{F}(P \wedge Q \wedge R))$ are the forecasts.

**COND** Similarly as for CONDCOND: we denote $(a, b, c) = (\mathbb{T}(P), \mathbb{T}(P \mid Q), \mathbb{T}(P \wedge Q))$ and the associated $(a', b', c')$ for the forecasts. Then we can compute

$$\begin{aligned}
&\mathbb{F}(P)\mathbb{F}(Q \mid P) - \mathbb{F}(P \wedge Q) \\
&= ab - c + \sigma\left(b\mathsf{N}(0, a(1-a)) + a\mathsf{N}(0, b(1-b)) - \mathsf{N}(0, c(1-c))\right) \\
&\quad + \sigma^2 \mathsf{N}(0, a(1-a))\mathsf{N}(0, b(1-b)).
\end{aligned}$$

The term next to $\sigma$ is a Gaussian with mean 0 and standard deviation:

$$\sigma\sqrt{a^2 b(1-b) + b^2 a(1-a) + c(1-c)} = \sigma\sqrt{ab\left(a(1-b) + b(1-a)\right) + c(1-c)}.$$

Again, we have to plug in $(a', b', c') = (\mathbb{F}(P), \mathbb{F}(Q \mid P), \mathbb{F}(P \wedge Q))$ instead of $(a, b, c)$.

Our violation metric is then:

$$v_{\text{COND}} = \frac{|a'b' - c'|}{\sqrt{a'b'\left(a'(1-b') + b'(1-a')\right) + c'(1-c') + \beta_{\text{MIN}}}}$$

And the test is again, for a suitable $\gamma$ corresponding to the desired power of the test:

$$v_{\text{COND}} > \gamma\sigma.$$

**PARAPHRASE** Here we can simply check whether $P$ and $Q$ are the same.

$$\begin{aligned}
\mathbb{F}(P) - \mathbb{F}(Q) &= \mathbb{T}(P) + \varepsilon_1 - \mathbb{T}(Q) - \varepsilon_2 \\
&= \varepsilon_1 - \varepsilon_2 \sim \mathsf{N}\left(0, \sigma^2((\mathbb{T}(P)(1 - \mathbb{T}(P)) + (\mathbb{T}(Q)(1 - \mathbb{T}(Q)))\right)
\end{aligned}$$

This yields the following violation metric:

$$v_{\text{PARAPHRASE}} = \frac{|\mathbb{F}(P) - \mathbb{F}(Q)|}{\sqrt{(\mathbb{F}(P)(1 - \mathbb{F}(P)) + (\mathbb{F}(Q)(1 - \mathbb{F}(Q)) + \beta_{\text{MIN}}}}$$

**ANDOR**

$$\begin{aligned}
&\mathbb{F}(P) + \mathbb{F}(Q) - \mathbb{F}(P \vee Q) - \mathbb{F}(P \wedge Q) \\
&= \mathbb{T}(P) + \mathbb{T}(Q) - \mathbb{T}(P \vee Q) - \mathbb{T}(P \wedge Q) + \varepsilon_1 + \varepsilon_2 - \varepsilon_3 - \varepsilon_4 \\
&= \varepsilon_1 + \varepsilon_2 - \varepsilon_3 - \varepsilon_4 \\
&\sim \mathsf{N}\left(0, \sigma^2\left(\mathbb{T}(P)(1 - \mathbb{T}(P)) + \mathbb{T}(Q)(1 - \mathbb{T}(Q))\right.\right. \\
&\quad \left.\left. + \mathbb{T}(P \vee Q)(1 - \mathbb{T}(P \vee Q)) + \mathbb{T}(P \wedge Q)(1 - \mathbb{T}(P \wedge Q))\right)\right).
\end{aligned}$$

We again plug in $\mathbb{F}$ instead of $\mathbb{T}$ to compute the error term allowed: $\gamma\sigma\sqrt{M}$ where

$$M = \mathbb{F}(P)(1 - \mathbb{F}(P)) + \mathbb{F}(Q)(1 - \mathbb{F}(Q) + \mathbb{F}(P \vee Q)(1 - \mathbb{F}(P \vee Q)) +$$
$$\mathbb{F}(P \wedge Q)(1 - \mathbb{F}(P \wedge Q))$$

and violation metric:

$$v_{\text{ANDOR}} = \frac{|\mathbb{F}(P) + \mathbb{F}(Q) - \mathbb{F}(P \vee Q) - \mathbb{F}(P \wedge Q)|}{\sqrt{\begin{array}{l} \mathbb{F}(P)(1 - \mathbb{F}(P)) + \mathbb{F}(Q)(1 - \mathbb{F}(Q)) + \\ \mathbb{F}(P \vee Q)(1 - \mathbb{F}(P \vee Q)) + \mathbb{F}(P \wedge Q)(1 - \mathbb{F}(P \wedge Q)) + \beta_{\text{MIN}} \end{array}}}.$$

**BUT**

$$\mathbb{F}(P \vee Q) - \mathbb{F}(P) - \mathbb{F}(\neg P \wedge Q) = \mathbb{T}(P \vee Q) - \mathbb{T}(P) - \mathbb{T}(\neg P \wedge Q) + \varepsilon_1 - \varepsilon_2 - \varepsilon_3 =$$
$$\varepsilon_1 - \varepsilon_2 - \varepsilon_3 \sim$$
$$\mathsf{N}\left(0, \sigma^2((\mathbb{T}(P \vee Q)(1 - \mathbb{T}(P \vee Q)) + (\mathbb{T}(P)(1 - \mathbb{T}(P)) + (\mathbb{T}(\neg P \wedge Q)(1 - \mathbb{T}(\neg P \wedge Q)))\right)$$

with error term:

$$\gamma \sigma \sqrt{\mathbb{F}(P \vee Q)(1 - \mathbb{F}(P \vee Q) + \mathbb{F}(P)(1 - \mathbb{F}(P) + \mathbb{F}(\neg P \wedge Q)(1 - \mathbb{F}(\neg P \wedge Q)}$$

and violation metric:

$$v_{\text{BUT}} = \frac{|\mathbb{F}(P \vee Q) - \mathbb{F}(P) - \mathbb{F}(\neg P \wedge Q)|}{\sqrt{\mathbb{F}(P \vee Q)(1 - \mathbb{F}(P \vee Q)) + \mathbb{F}(P)(1 - \mathbb{F}(P)) + \mathbb{F}(\neg P \wedge Q)(1 - \mathbb{F}(\neg P \wedge Q) + \beta_{\text{MIN}}}}$$

**CONSEQUENCE** In the case of inequalities involving $\leq$, there are two ways in which the consistency check can be passed. If $\mathbb{F}(P) \leq \mathbb{F}(Q)$, the consistency check is automatically passed. Otherwise, we check for pseudo-equality using the same violation metric as in PARAPHRASE.

$$v_{\text{CONSEQUENCE}} = [\mathbb{F}(P) > \mathbb{F}(Q)] \frac{|\mathbb{F}(P) - \mathbb{F}(Q)|}{\sqrt{\mathbb{F}(P)(1 - \mathbb{F}(P)) + \mathbb{F}(Q)(1 - \mathbb{F}(Q)) + \beta_{\text{MIN}}}}$$

where $[\mathbb{F}(P) > \mathbb{F}(Q)]$ is the Iverson Bracket (1 if true, 0 otherwise).

**AND** Similarly to CONSEQUENCE, if the chain of strict inequalities

$$\max(\mathbb{F}(P) + \mathbb{F}(Q) - 1, 0) < \mathbb{F}(P \wedge Q) < \min(\mathbb{F}(P), \mathbb{F}(Q))$$

holds, then the check automatically passes. We set $v_{\text{AND\_LHS}} = 0$ and $v_{\text{AND\_RHS}} = 0$ if it passes the first and second strict inequality respectively.

If not, then we test for pseudo-equality for the violating pair:

LHS : $\max(\mathbb{F}(P) + \mathbb{F}(Q) - 1, 0) = \mathbb{F}(P \wedge Q)$

RHS : $\mathbb{F}(P \wedge Q) = \min(\mathbb{F}(P), \mathbb{F}(Q))$

Equality check if it fails the first inequality:

$$\varepsilon_{\text{LHS}} = \begin{cases} \gamma \sigma \sqrt{\mathbb{F}(P)(1 - \mathbb{F}(P)) + \mathbb{F}(Q)(1 - \mathbb{F}(Q)) + \mathbb{F}(P \wedge Q)(1 - \mathbb{F}(P \wedge Q))} \\ \quad \text{if } \mathbb{F}(P) + \mathbb{F}(Q) - 1 > 0, \\ \\ \text{N/A} \\ \quad \text{otherwise pass as } \mathbb{F}(P \wedge Q) \geq 0. \end{cases}$$

$$v_{\text{AND\_LHS}} = [\mathbb{F}(P) + \mathbb{F}(Q) - 1 > \mathbb{F}(P \wedge Q)] \cdot$$
$$\frac{\mathbb{F}(P) + \mathbb{F}(Q) - 1 - \mathbb{F}(P \wedge Q)}{\sqrt{\mathbb{F}(P)(1 - \mathbb{F}(P)) + \mathbb{F}(Q)(1 - \mathbb{F}(Q)) + \mathbb{F}(P \wedge Q)(1 - \mathbb{F}(P \wedge Q)) + \beta_{\text{MIN}}}}$$

Equality check if it fails the second inequality:

Define $\mathbb{F}(R) = \min(\mathbb{F}(P), \mathbb{F}(Q))$.

$$\varepsilon_{\text{RHS}} = \gamma \sigma \sqrt{\mathbb{F}(P \wedge Q)(1 - \mathbb{F}(P \wedge Q)) + \mathbb{F}(R)(1 + \mathbb{F}(R))}$$
$$v_{\text{AND\_RHS}} = [\mathbb{F}(R) < \mathbb{F}(P \wedge Q)] \frac{\mathbb{F}(P \wedge Q) - \mathbb{F}(R)}{\sqrt{\mathbb{F}(P \wedge Q)(1 - \mathbb{F}(P \wedge Q)) + \mathbb{F}(R)(1 - \mathbb{F}(R)) + \beta_{\text{MIN}}}}$$

Consistency is violated if either inequality is violated, *and* the respective hypothesis test for pseudo-equality fails. We use $v_{\text{AND\_LHS}}$ for the first and $v_{\text{AND\_RHS}}$ for the second inequality. We define $v_{\text{AND}} = \max\{v_{\text{AND\_LHS}}, v_{\text{AND\_RHS}}\}$.

**OR**   We proceed similarly as for AND.

If the strict inequality $\max(\mathbb{F}(P), \mathbb{F}(Q)) < \mathbb{F}(P \vee Q) < \min(1, \mathbb{F}(P) + \mathbb{F}(Q))$ holds, then it automatically passes. We set $v_{\text{OR\_LHS}} = 0$ and $v_{\text{OR\_RHS}} = 0$ if it passes the first and second strict inequality respectively.

If not, we test for pseudo-equality:

LHS : $\max(\mathbb{F}(P), \mathbb{F}(Q)) = \mathbb{F}(P \vee Q)$

RHS : $\mathbb{F}(P \vee Q) = \min(1, \mathbb{F}(P) + \mathbb{F}(Q))$.

Equality check LHS: Define $\mathbb{F}(S) = \max(\mathbb{F}(P), \mathbb{F}(Q))$.

$$\varepsilon_{\text{LHS}} = \gamma \sigma \sqrt{\mathbb{F}(S)(1 - \mathbb{F}(S)) + \mathbb{F}(P \vee Q)(1 - \mathbb{F}(P \vee Q))}$$

$$v_{\text{OR\_LHS}} = [\mathbb{F}(S) > \mathbb{F}(P \vee Q)] \frac{\mathbb{F}(S) - \mathbb{F}(P \vee Q)}{\sqrt{\mathbb{F}(S)(1 - \mathbb{F}(S)) + \mathbb{F}(P \vee Q)(1 - \mathbb{F}(P \vee Q)) + \beta_{\text{MIN}}}}$$

Equality check RHS:

$$\varepsilon_{\text{RHS}} = \begin{cases} \gamma \sigma \sqrt{\mathbb{F}(P \vee Q)(1 - \mathbb{F}(P \vee Q)) + \mathbb{F}(P)(1 - \mathbb{F}(P)) + \mathbb{F}(Q)(1 - \mathbb{F}(Q))} \\ \quad \text{if } \mathbb{F}(P) + \mathbb{F}(Q) < 1, \\ \text{N/A} \\ \quad \text{otherwise pass as } \mathbb{F}(P \vee Q) \le 1. \end{cases}$$

$$v_{\text{OR\_RHS}} = [\mathbb{F}(P) + \mathbb{F}(Q) < \mathbb{F}(P \vee Q)] \cdot$$
$$\frac{\mathbb{F}(P \vee Q) - \mathbb{F}(P) - \mathbb{F}(Q)}{\sqrt{\mathbb{F}(P \vee Q)(1 - \mathbb{F}(P \vee Q)) + \mathbb{F}(P)(1 - \mathbb{F}(P)) + \mathbb{F}(Q)(1 - \mathbb{F}(Q)) + \beta_{\text{MIN}}}}$$

Consistency is violated if either inequality is violated, *and* the subsequent hypothesis test for pseudo-equality fails. We use $v_{\text{OR\_LHS}}$ for the first and $v_{\text{OR\_RHS}}$ for the second inequality. Analogously to AND, define $v_{\text{OR}} = \max\{v_{\text{OR\_LHS}}, v_{\text{OR\_RHS}}\}$.

**EXPEVIDENCE**  Write $(a, b, c, d) = (\mathbb{T}(P), \mathbb{T}(P \mid Q), \mathbb{T}(P \mid \neg Q), \mathbb{T}(Q))$; then

$$
\begin{aligned}
b'&d' + c'(1 - d') - a' \\
&= (b + \sigma N(b(1 - b)))(d + \sigma N(d(1 - d))) \\
&\quad + (c + \sigma N(c(1 - c)))(1 - d - \sigma N(d(1 - d))) \\
&\quad - (a + \sigma N(a(1 - a))) \\
&= (bd + c(1 - d) - a) \\
&\quad + \sigma \left[ dN(b(1 - b)) \right. \\
&\quad\quad + (b - c)N(d(1 - d)) \\
&\quad\quad + (1 - d)N(c(1 - c)) \\
&\quad\quad \left. - N(a(1 - a)) \right] \\
&\quad + O(\sigma^2)
\end{aligned}
$$

gives us a normal distribution with standard deviation

$$
\sigma\sqrt{a(1 - a) + d^2 b(1 - b) + (1 - d)^2 c(1 - c) + (b - c)^2 d(1 - d)}.
$$

The violation metric is then:

$$
\frac{|bd + c(1 - d) - a|}{\sqrt{a(1 - a) + d^2 b(1 - b) + (1 - d)^2 c(1 - c) + (b - c)^2 d(1 - d) + \beta_{\text{MIN}}}}.
$$

**Hyperparameters for hypothesis testing**  Our goal is for the rejection criteria to be similar to the arbitrage violation metric in Appendix D on simple examples. We choose $\gamma = 2.58$ for all checks, to ensure 99%-confidence intervals for two-sided tests; future work may consider using a different $\gamma$ for checks that require one-sided tests. We pick $\sigma = 0.05$ (corresponding to $n = 400$ in Definition E.1). The allowed violation threshold for all checks is then $\gamma\sigma = 0.129$. For reference, a NEGATION pair $(\mathbb{F}(P), \mathbb{F}(\neg P)) = (0.5, 0.59)$ has a violation metric of 0.128, and would thus not be rejected as inconsistent. This closely corresponds to the tolerance threshold of $10^{-2}$ of profit for the arbitrage metric, described in Section 2.1.

We pick $\beta_{\text{MIN}} = 10^{-3}$ because LLM forecasters from Halawi et al. (2024) answer with at most 3 digits of precision for events close to 0 and 1 in probability.

# F    FORECASTERS

We describe the forecaster architectures evaluated in the paper below. All of these forecasters accept a model parameter working with most popular LLMs, such as `gpt-4o`, `claude-3.5-sonnet` and `llama-3.1-405B`.

In plots, the following names refer to these forecasters:

- `GPT-4o-05`: Basic Forecaster with `gpt-4o-2024-05-13`
- `GPT-4o-08`: Basic Forecaster with `gpt-4o-2024-08-06`
- `GPT-4o-mini`: Basic Forecaster with `gpt-4o-mini-2024-07-18`
- `Sonnet`: Basic Forecaster with `claude-3.5-sonnet`
- `L3-8B`: Basic Forecaster with `llama-3.1-8B`
- `L3-70B`: Basic Forecaster with `llama-3.1-70B`
- `L3-405B`: Basic Forecaster with `llama-3.1-405B`
- `CoT-o1-preview`: CoT Forecaster with `o1-preview`
- `CoT-o1-mini`: CoT Forecaster with `o1-mini`
- `CoT-GPT-4o-08`: CoT Forecaster with `gpt-4o-2024-08-06`
- `CoT-GPT-4o-mini`: CoT Forecaster with `gpt-4o-mini`
- `CoT-Sonnet`: CoT Forecaster with `claude-3.5-sonnet`
- `CoT-L3-8B`: CoT Forecaster with `llama-3.1-8B`
- `CoT-L3-70B`: CoT Forecaster with `llama-3.1-70B`
- `CoT-L3-405B`: CoT Forecaster with `llama-3.1-405B`

All forecasters receive the question (see Appendix C.1) as a string render of the JSON object in Figure 11.

```
{
"title": "Question title",
"body": "Question body and resolution criteria",
"resolution_date": "YYYY-MM-DD",
"created_date": "YYYY-MM-DD"
}
```

Figure 11: The format in which questions are presented to forecasters. If `created_date` is not available, it is omitted.

## F.1    BASIC FORECASTER

The Basic Forecaster is a simple forecasting model that uses a language model to generate probability estimates for given questions. We use the Instructor library Liu (2024) to make the output conform to a specific Pydantic model that has a `prob` field forced to be a float between 0 and 1.

> You are an informed and well-calibrated forecaster. I need you to give me your best probability estimate for the following sentence or question resolving YES. Your answer should be a float between 0 and 1, with nothing else in your response. Question: {question}

Figure 12: The prompt used for Basic Forecaster.

## F.2 CoT Forecaster

The CoTForecaster is composed of two steps:

1. The first model call is a native chat message with a chain-of-thought reasoning prompt in Figure 13.
2. Then, `gpt-4o-mini` is used in an Instructor Liu (2024) call to parse the output into a single probability estimate similarly as in the Basic Forecaster, plus the reasoning summary.

We use this two-step process because of concerns with structured outputs degrading reasoning ability in language models.

---

You are an informed and well-calibrated forecaster. I need you to give me your best probability estimate for the following question resolving YES. If you think it is likely the question resolves YES, the probability should be large; if you think it is unlikely the question resolves NO, the probability should be small. I want you to first provide a detailed reasoning for your answer, and then give me the probability. Your answer should be in the format: 'Reasoning: [your reasoning here] Probability: [float between 0 and 1]'

Note: unless explicitly stated in the prompt, do not worry about the exact formatting of the output. There will be an extra step that will summarize your output into the final answer format. For context, the final answer format is described by the following Pydantic model: {`response_model.model_fields=`} Again, just try to answer the question as best as you can, with all the necessary information; the output will be cleaned up in the final step. Question: {question}

---

Figure 13: The prompt used for CoT Forecaster.

---

**Algorithm 1** `ArbitrageForecaster` algorithm: $\langle \mathbb{F} \rangle_{\vec{C}}$

---

    **input** $x$
    $p \leftarrow \mathbb{F}(x)$                                                 $\triangleright$ Query base forecaster
    $w \leftarrow 1$
    **for** $(\mathcal{R}_i, \mathcal{S}_i, \mathcal{J}_i)$ in $\vec{C}$ **do**
        $(x, x_2, \ldots x_n) \leftarrow \mathcal{J}_i(x)$                     $\triangleright$ Instantiate tuple of size $n = n_{\mathcal{R}_i}$
        $(p_2, \ldots p_n) \leftarrow (\mathbb{F}(x_2), \ldots \mathbb{F}(x_n))$          $\triangleright$ Query base forecaster on tuple
        $(p, p_2, \ldots p_n) \leftarrow \mathcal{A}_i^{(w,1,\ldots 1)}(p, p_2, \ldots p_n)$     $\triangleright$ arbitrage the forecasts as per Def 2
        $w \leftarrow w + n - 1$              $\triangleright$ $p$ now carries information from $n - 1$ other markets
    **end for**
    **return** p

---

## G   ARBITRAGEFORECASTER

To formally define `ArbitrageForecaster`, we need to first formalize our "instantiation" process mathematically:

**Definition G.1** (Tuple sampler). Let $\mathcal{R} : \text{Prop}^n \to \{\top, \bot\}$, $\mathcal{S} : \Delta\Theta^n \to \{\top, \bot\}$ be a consistency check. Then we call $\mathcal{J} : \text{Prop} \rightsquigarrow \text{Prop}^n$ a "single-base-question tuple sampler" for $\mathcal{R}$ if for all $x$, $\mathcal{J}(x)_1 = x$ and $\mathcal{R}(\mathcal{J}(x))$ holds surely.

A multiple-base-question tuple sampler $\mathcal{I} : \text{Prop}^m \to \text{Prop}^n$, like the instantiation process described in 3.2, can simply be composed with a question sampler $\mathcal{G} : \text{Prop} \rightsquigarrow \text{Prop}$ (e.g a synthetic generator or a sampler from our dataset) to produce a single-base-question sampler $\mathcal{J}(x) := \mathcal{I}(x, \mathcal{G}(x), \ldots \mathcal{G}(x))$.

Next, in order to correctly handle sequentially arbitraging checks and prevent bias towards later applied checks, we need to introduce "weighted" arbitraging. This follows easily from Eq D.1 by simply having the scoring rule for each question $x$ be $w_x \log(p)$. We denote the calculation of arbitraged probabilities under these weighted scoring rules by $\mathcal{A}^{(w_1, \ldots w_n)}$.

**Definition G.2** (`ArbitrageForecaster`). Let $\mathbb{F} : \text{Prop} \to \Delta\Theta$ be the "Base Forecaster", and let $\vec{C} := [(\mathcal{R}_1, \mathcal{S}_1, \mathcal{J}_1), \ldots (\mathcal{R}_k, \mathcal{S}_k, \mathcal{J}_k)]$ be a list of consistency checks along with respective single-base-question tuple samplers. Then we construct a new forecaster $\langle \mathbb{F} \rangle_{\vec{C}} : \text{Prop} \to \Delta\Theta$ that produces its forecast for a given question $x$ as given in Algorithm 1; we call this the `ArbitrageForecaster` with base $\mathbb{F}$ and check list $\vec{C}$.

The first thing we observe is that this isn't necessarily *robust* to different instantiations. For this reason, we a priori expect that **ArbitrageForecaster will be more effective on** single-base-question checks like NEGATION and PARAPHRASE.

We might hope that the `ArbitrageForecaster` introduced in Def G.2 would be definitionally consistent on the checks it is arbitraged on. However, this is not the case *even for* `ArbitrageForecaster` *applied to a single check* $\mathcal{R}(x_1, \ldots x_n)$, because the tuple of forecasts that is arbitraged to compute $\langle \mathbb{F} \rangle_{(\mathcal{R}, \mathcal{S}, \mathcal{J})}(x_1)$, the tuple arbitraged to compute $\langle \mathbb{F} \rangle_{(\mathcal{R}, \mathcal{S}, \mathcal{J})}(x_2)$, ..., the tuple arbitraged to compute $\langle \mathbb{F} \rangle_{(\mathcal{R}, \mathcal{S}, \mathcal{J})}(x_n)$ are all different. While the tuple instantiated to compute $\langle \mathbb{F} \rangle_{(\mathcal{R}, \mathcal{S}, \mathcal{J})}(x_1)$ could indeed be $\mathcal{J}(x_1) = (x_1, \ldots x_n)$ (at least if the tuple sampler $\mathcal{J}$ is deterministic and happens to be the same as the one used in the instantiation of the check), the tuples instantiated to compute $\langle \mathbb{F} \rangle_{(\mathcal{R}, \mathcal{S}, \mathcal{J})}(x_i)$ for $i \neq 1$ will be $\mathcal{J}(x_i)$, all of which are different from one another.

To make this concrete, consider the simplest case of $\langle \mathbb{F} \rangle_P$ (where $P$ is short for PARAPHRASE); let para be a deterministic tuple-sampler for PARAPHRASE. $\langle \mathbb{F} \rangle_P(x)$ is calculated by arbitraging $\mathbb{F}(x)$ and $\mathbb{F}(\text{para}(x))$. But $\mathbb{F}(\text{para}(x))$ is calculated by arbitraging $\mathbb{F}(\text{para}(x))$ and $\mathbb{F}(\text{para}(\text{para}(x)))$.

A priori, this gives us the following hypothesis: **ArbitrageForecaster will be especially effective for fundamentally "symmetric" checks like NEGATION** – where $\text{neg}(\text{neg}(P))$ is likely to be a very similar sentence to $P$. Although we have not conducted a full scale experiment of `ArbitrageForecaster` with each checker, our preliminary results in Table 4 do suggest very good performance of `ArbitrageForecaster` on NEGATION.

Suppose, however, that we had an "extended" `ArbitrageForecaster` that made its forecast for $x$ based on the tuple $(x, \text{para}(x), \text{para}^2(x), \ldots \text{para}^r(x))$ – then its forecast for $\text{para}(x)$ would be based on $(\text{para}(x), \text{para}^2(x), \ldots \text{para}^{r+1}(x))$ – these tuples would be "almost" the same, except with $\text{para}^{r+1}(x)$ instead of $x$, and this extended `ArbitrageForecaster` would be "almost" consistent on PARAPHRASE.

This is precisely the idea behind recursively applying `ArbitrageForecaster` to itself: we recursively define $\langle \mathbb{F} \rangle^r(x) := \mathcal{A}(\langle \mathbb{F} \rangle^{r-1}(\mathcal{J}(x)_i)$ for $i = 1, \ldots n)$ – then *if* this iteration approaches a fixed point, this fixed point $\langle \mathbb{F} \rangle^{\infty}$ is consistent. More precisely:

**Theorem G.3** (Consistency of recursive `ArbitrageForecaster`). *Let $(\mathcal{R}, \mathcal{S}, \mathcal{J})$ be an n-ary consistency check and a corresponding deterministic tuple sampler satisfying Def G.1, and have $\mathcal{A}(p_1, \ldots p_n)$ and $\mathcal{V}(p_1, \ldots p_n)$ denote the arbitraging function and arbitrage metric corresponding to $\mathcal{R}$ as per Def D.1 under a logarithmic scoring rule. Then, for some "base forecaster" $\langle \mathbb{F} \rangle^0 = \mathbb{F}$, recursively define*

$$\langle \mathbb{F} \rangle^r(x) := \mathcal{A}(\langle \mathbb{F} \rangle^{r-1}(\mathcal{J}(x)_i) \text{ for } i = 1, \ldots n)$$

*If this iteration converges pointwise in log-odds space – i.e. if for all $x \in \text{Prop}$, the sequence $\langle \mathbb{F} \rangle^r(x)$ has a limit strictly between 0 and 1, then $\mathcal{V}(\langle \mathbb{F} \rangle^r(\mathcal{J}(x)_i)$ for $i = 1, \ldots n) \to 0$.*

*Proof.* Recall as per Def D.1 that, where $\Omega$ is the set of possible outcomes allowed by $\mathcal{R}$:

$$\mathcal{V}(\langle \mathbb{F} \rangle^r(\mathcal{J}(x)_i) \text{ for } i = 1, \ldots n)$$

$$= \min_{\omega \in \Omega} \sum_{i=1}^{n} \left( \log(\mathcal{A}(\langle \mathbb{F} \rangle^r(\mathcal{J}(x)_j) \text{ for } j = 1, \ldots n)_i) - \log\langle \mathbb{F} \rangle^r(\mathcal{J}(x)_i) \right) \delta_{\omega(i) = \top}$$

$$+ \left( \log(1 - \mathcal{A}(\langle \mathbb{F} \rangle^r(\mathcal{J}(x)_j) \text{ for } j = 1, \ldots n)_i) - \log(1 - \langle \mathbb{F} \rangle^r(\mathcal{J}(x)_i)) \right) \delta_{\omega(i) = \bot}$$

$$= \min_{\omega \in \Omega} \sum_{i=1}^{n} \left( \log\langle \mathbb{F} \rangle^{r+1}(\mathcal{J}(x)_i)) - \log\langle \mathbb{F} \rangle^r(\mathcal{J}(x)_i) \right) \delta_{\omega(i) = \top}$$

$$+ \left( \log(1 - \langle \mathbb{F} \rangle^{r+1}(\mathcal{J}(x)_i) - \log(1 - \langle \mathbb{F} \rangle^r(\mathcal{J}(x)_i)) \right) \delta_{\omega(i) = \bot}$$

Since $\langle \mathbb{F} \rangle^r(x)$ converges to something that is neither 0 nor 1, so do $\log\langle \mathbb{F} \rangle^r(x)$ and $\log(1 - \langle \mathbb{F} \rangle^r(x))$. And as this is true for *all* $x$, so in particular it is true for $\mathcal{J}(x)_i$. Thus the expression above is a finite sum of terms that each approach 0. $\square$

This is a somewhat weak result: other than for NEGATION and PARAPHRASE, none of our static consistency checks involved a deterministic instantiation process – they all require sampling other related base questions, and having the checks use the same instantiation process as the `ArbitrageForecaster` would be cheating.

Furthermore, this gives us no actual conditions for the convergence of the iteration. At least for PARAPHRASE, we have the following – where $\log \text{odds} \, p$ denotes $\log \frac{p}{1-p}$:

**Theorem G.4** (Convergence of recursive `ArbitrageForecaster` for PARAPHRASE). *If the sequence $a_i = \log \text{odds} \, \mathbb{F}(\text{para}^i(x))$ is convergent, then the condition of Theorem G.3 holds for the recursive `ArbitrageForecaster` defined arbitraged on PARAPHRASE with tuple sampler para.*

*Proof.* Recall from Sec D.1 that the arbitraged probability for PARAPHRASE is simply the average of the original probabilities in log-odds space, i.e. $\log \text{odds} \, \mathcal{A}(\mathbb{F}(x), \mathbb{F}(\text{para}(x))) = \frac{\log \text{odds} \, \mathbb{F}(x) + \log \text{odds} \, \mathbb{F}(\text{para}(x))}{2}$. We can apply this recursively to get:

$$\langle \mathbb{F} \rangle^r(x) = \frac{1}{2^r} \sum_{i=0}^{r} \binom{r}{i} \log \text{odds} \, \mathbb{F}(\text{para}^i(x))$$

Which is simply a binomial moving average of $\log \text{odds} \, \mathbb{F}(\text{para}^i(x)) = a_i$, and converges iff $a_i$ does. Convergence in log-odds space is equivalent to convergence of probability to something other than 0 or 1, so the result follows. $\square$

### G.1 Choices of experiments

A single call to $\langle \mathbb{F} \rangle_{\vec{C}}$, where $\vec{C} := [(\mathcal{R}_1, \mathcal{S}_1, \mathcal{J}_1), ...(\mathcal{R}_k, \mathcal{S}_k, \mathcal{J}_k)]$, involves $1 + \sum_i (n_{\mathcal{R}_i} - 1)$ calls to $\mathbb{F}$, plus at least $\sum_i (m_{\mathcal{R}_i} + n_{\mathcal{R}_i} - 2)$ (where $m_{\mathcal{R}_i}$ is the number of separate base questions that must be generated synthetically in each tuple) LLM calls for the $\mathcal{J}_i$s.

For all the checks listed in Table 3, this amounts to a total of 49 LLM calls per question. For a *recursive* ArbitrageForecaster set-up of depth $r$, this amounts to $49^r$ LLM calls per question, which can get prohibitively expensive. Even on gpt-4o-mini and assuming $\approx 600$ input tokens and 600 output tokens on average, this amounts to $\approx \$0.02$ per question at depth $r = 1$, and $\approx \$2500$ per question at depth $r = 4$.

Furthermore, it was not clear that experimenting on all checks made logical sense: recursive ArbitrageForecaster set-ups with COND, CONDCOND and EXPEVIDENCE would involve forms like $P \mid (Q \mid R)$, which do not have a basis in probability theory. We decided to prioritize studying the following hypotheses and research questions, motivated by the theoretical discussion above:

1. We hypothesised above that ArbitrageForecaster will be **particularly effective on checks that are symmetric and have deterministic instantiations** – thus we studied $\langle$gpt-4o-mini$\rangle_{\text{NEGATION}}$.

2. We hypothesized that there would be **consistency gains from increasing depth** $r$ – thus we studied recursive ArbitrageForecaster setups on NEGATION an PARAPHRASE, where it was most practical to.

3. We were interested to know **if the consistency gains observed when arbitraging on one check alone would persist after arbitraging on a sequence of checks** – to predict if this would hold when arbitraging on the full sequence of checks, we did a preliminary run of $\langle$gpt-4o-mini$\rangle_{\text{NEGATION,PARAPHRASE}}$ and tested if it maintains consistency on NEGATION and PARAPHRASE.

4. **We expected** $\langle \mathbb{F} \rangle_{\text{EXPEVIDENCE}}$ **to improve ground truth and consistency scores across the board.** This is based on our intuition that arbitraging on EXPEVIDENCE essentially "informs" the forecast on a question $x$ with consideration information $y$ – except instead of subjectively feeding this information (e.g. in chain-of-thought), it adjusts for it via a strict probabilistic rule. Although a recursive setup would not make sense for EXPEVIDENCE, $\langle \mathbb{F} \rangle_{[\text{EXPEVIDENCE}]*r}$ simply sequentially arbitrages on EXPEVIDENCE repeatedly (breaking the seed each time to ensure unique new questions $y$), which amounts to informing the forecast for $x$ with information $y_1$, $y_2$ etc.

The results reported in Sec 5 of the main body and G.2 of the Appendix provide evidence in favour of hypotheses 1 and 2, answer 3 in the affirmative, and do not provide clear evidence on 4.

Future work should compare $\langle \mathbb{F} \rangle_{[\text{EXPEVIDENCE}]}$ against a comparable chain-of-thought model in which the forecaster is asked to consider these related questions before it makes its forecast.

### G.2 Results tables for ArbitrageForecaster

Consistency violation and ground truth results for each of the ArbitrageForecaster configurations we experimented with are reported in Tables 4, 5, 6 and 7. The results included are for the NewsAPI dataset and the arbitrage metric. Results for the scraped and 2028 synthetic datasets (Appendix L), as well as for the frequentist metric, look very similar; they are available in the supplementary data of this paper.

Table 4: Consistency results (arbitrage metric) for $\langle\texttt{gpt-4o-mini}\rangle^r_{\text{NEGATION}}$ (denoted CF-Nr) forecasters on NewsAPI questions.

| Check | gpt-4o-mini | | CF-N1 | | CF-N2 | | CF-N3 | | CF-N4 | |
|---|---|---|---|---|---|---|---|---|---|---|
| | Avg | Frac | Avg | Frac | Avg | Frac | Avg | Frac | Avg | Frac |
| NEGATION | 0.036 | 43% | 0.012 | 33% | 0.007 | 22% | 0.004 | 11% | 0.004 | 9% |
| PARAPHRASE | 0.013 | 27% | 0.012 | 36% | 0.008 | 23% | 0.006 | 16% | 0.005 | 17% |
| CONDCOND | 0.084 | 85% | 0.111 | 88% | 0.121 | 91% | 0.129 | 94% | 0.136 | 93% |
| EXPEVIDENCE | 0.015 | 27% | 0.009 | 35% | 0.008 | 25% | 0.007 | 26% | 0.007 | 25% |
| CONSEQUENCE | 0.005 | 10% | 0.003 | 9% | 0.003 | 7% | 0.002 | 4% | 0.001 | 3% |
| AND | 0.006 | 20% | 0.019 | 45% | 0.027 | 53% | 0.031 | 59% | 0.035 | 65% |
| OR | 0.007 | 13% | 0.004 | 10% | 0.002 | 6% | 0.002 | 6% | 0.001 | 4% |
| ANDOR | 0.017 | 38% | 0.024 | 58% | 0.031 | 61% | 0.033 | 67% | 0.035 | 66% |
| BUT | 0.053 | 75% | 0.081 | 84% | 0.091 | 89% | 0.100 | 88% | 0.107 | 91% |
| COND | 0.062 | 88% | 0.085 | 92% | 0.107 | 91% | 0.119 | 94% | 0.131 | 96% |
| aggregated | 0.030 | | 0.036 | | 0.041 | | 0.043 | | 0.046 | |
| Brier score | 0.185 | | 0.204 | | 0.202 | | 0.201 | | 0.201 | |

Table 5: Consistency results (arbitrage metric) for $\langle\texttt{gpt-4o-mini}\rangle^r_{\text{PARAPHRASE}}$ (denoted CF-Pr) forecasters on NewsAPI questions.

| Check | gpt-4o-mini | | CF-P1 | | CF-P2 | | CF-P3 | | CF-P4 | |
|---|---|---|---|---|---|---|---|---|---|---|
| | Avg | Frac | Avg | Frac | Avg | Frac | Avg | Frac | Avg | Frac |
| NEGATION | 0.036 | 43% | 0.028 | 49% | 0.026 | 50% | 0.023 | 46% | 0.024 | 44% |
| PARAPHRASE | 0.013 | 27% | 0.006 | 22% | 0.004 | 11% | 0.002 | 6% | 0.002 | 3% |
| CONDCOND | 0.084 | 85% | 0.083 | 83% | 0.079 | 85% | 0.080 | 83% | 0.079 | 84% |
| EXPEVIDENCE | 0.015 | 27% | 0.014 | 28% | 0.012 | 24% | 0.011 | 28% | 0.012 | 28% |
| CONSEQUENCE | 0.005 | 10% | 0.002 | 4% | 0.001 | 3% | 0.001 | 2% | 0.001 | 2% |
| AND | 0.006 | 20% | 0.004 | 12% | 0.005 | 13% | 0.004 | 12% | 0.004 | 12% |
| OR | 0.007 | 13% | 0.005 | 10% | 0.004 | 9% | 0.003 | 10% | 0.003 | 9% |
| ANDOR | 0.017 | 38% | 0.015 | 41% | 0.014 | 42% | 0.013 | 39% | 0.013 | 39% |
| BUT | 0.053 | 75% | 0.053 | 76% | 0.049 | 77% | 0.051 | 79% | 0.048 | 79% |
| COND | 0.062 | 88% | 0.066 | 93% | 0.071 | 95% | 0.069 | 95% | 0.071 | 95% |
| aggregated | 0.030 | | 0.028 | | 0.026 | | 0.026 | | 0.026 | |
| Brier score | 0.185 | | 0.176 | | 0.175 | | 0.174 | | 0.175 | |

Table 6: Consistency results (arbitrage metric) for $\langle\texttt{gpt-4o-mini}\rangle^r_{[\text{NEGATION},\text{PARAPHRASE}]}$ (denoted CF-NPr) forecasters on NewsAPI questions.

| Check | gpt-4o-mini | | CF-NP1 | | CF-NP2 | | CF-NP3 | | CF-NP4 | |
|---|---|---|---|---|---|---|---|---|---|---|
| | Avg | Frac | Avg | Frac | Avg | Frac | Avg | Frac | Avg | Frac |
| NEGATION | 0.036 | 43% | 0.014 | 30% | 0.007 | 18% | 0.004 | 9% | 0.003 | 6% |
| PARAPHRASE | 0.013 | 27% | 0.006 | 17% | 0.003 | 7% | 0.002 | 2% | 0.001 | 2% |
| CONDCOND | 0.084 | 85% | 0.095 | 90% | 0.096 | 86% | 0.108 | 94% | 0.115 | 94% |
| EXPEVIDENCE | 0.015 | 27% | 0.010 | 27% | 0.007 | 27% | 0.006 | 22% | 0.005 | 21% |
| CONSEQUENCE | 0.005 | 10% | 0.003 | 7% | 0.001 | 3% | 0.001 | 2% | 0.001 | 0% |
| AND | 0.006 | 20% | 0.011 | 30% | 0.010 | 28% | 0.011 | 34% | 0.012 | 39% |
| OR | 0.007 | 13% | 0.004 | 11% | 0.002 | 5% | 0.001 | 4% | 0.001 | 2% |
| ANDOR | 0.017 | 38% | 0.017 | 43% | 0.016 | 46% | 0.016 | 46% | 0.016 | 47% |
| BUT | 0.053 | 75% | 0.070 | 85% | 0.072 | 91% | 0.077 | 91% | 0.083 | 97% |
| COND | 0.062 | 88% | 0.082 | 96% | 0.076 | 97% | 0.076 | 97% | 0.077 | 98% |
| aggregated | 0.030 | | 0.031 | | 0.029 | | 0.030 | | 0.031 | |
| Brier score | 0.185 | | 0.188 | | 0.195 | | 0.200 | | 0.202 | |

Table 7: Consistency results (arbitrage metric) for $\langle \texttt{gpt-4o-mini} \rangle_{[\text{ExpEvidence}]*r}$ (denoted CF-rxEE1) forecasters on NewsAPI questions.

| Check | gpt-4o-mini | | CF-1xEE1 | | CF-2xEE1 | | CF-3xEE1 | | CF-4xEE1 | |
|---|---|---|---|---|---|---|---|---|---|---|
| | Avg | Frac | Avg | Frac | Avg | Frac | Avg | Frac | Avg | Frac |
| NEGATION | 0.036 | 43% | 0.030 | 51% | 0.026 | 49% | 0.024 | 50% | 0.025 | 53% |
| PARAPHRASE | 0.013 | 27% | 0.008 | 22% | 0.006 | 22% | 0.005 | 19% | 0.005 | 18% |
| CONDCOND | 0.084 | 85% | 0.057 | 82% | 0.053 | 79% | 0.050 | 76% | 0.044 | 74% |
| EXPEVIDENCE | 0.015 | 27% | 0.008 | 22% | 0.007 | 19% | 0.007 | 16% | 0.007 | 20% |
| CONSEQUENCE | 0.005 | 10% | 0.003 | 8% | 0.002 | 7% | 0.002 | 5% | 0.002 | 6% |
| AND | 0.006 | 20% | 0.002 | 6% | 0.002 | 6% | 0.002 | 4% | 0.001 | 5% |
| OR | 0.007 | 13% | 0.004 | 9% | 0.003 | 8% | 0.002 | 8% | 0.003 | 9% |
| ANDOR | 0.017 | 38% | 0.014 | 42% | 0.011 | 39% | 0.010 | 34% | 0.011 | 35% |
| BUT | 0.053 | 75% | 0.040 | 71% | 0.039 | 74% | 0.040 | 77% | 0.035 | 68% |
| COND | 0.062 | 88% | 0.049 | 88% | 0.046 | 89% | 0.044 | 88% | 0.040 | 87% |
| aggregated | 0.030 | | 0.021 | | 0.020 | | 0.019 | | 0.017 | |
| Brier score | 0.185 | | 0.172 | | 0.171 | | 0.171 | | 0.173 | |

## H    PROMPTS FOR THE EVALUATION PIPELINE

In this section, we present the prompts used for the different parts of our pipeline. For each LLM call, we use `gpt-4o` with a structured output Pydantic format enforced by the Instructor library Liu (2024) and JSON API calls. **The whitespace in the figures is not representative of the whitespace in actual queries.**

---

**Synthetic question generation prompt**

I want you to help me generate some forecasting questions for a forecasting market site like Metaculus or PredictIt. I will provide you with a category and some tags. Your task is to generate questions that can be answered with a probability between 0 and 1. For each tag, generate a relevant question if the tag is pertinent to the category. If the tag is not relevant, generate a general question about the category.
Examples:
{example_1}
{example_2}
{example_3}
{example_4}
{example_5}
{example_6}
Category: {category} Tags: {tags}

---

Figure 14: The prompt used for generating the *title* field of forecasting questions, given the *category* and *tags* metadata.

A list of initial quality-filtered questions is supplied to seed the list of examples.

---

**Relevance scoring prompt**

I'm doing a project that involve eliciting probabilities from LLMs to measure the calibration, consistency and such properties of LLM forecasters. As part of this project we will be taking logical combinations of forecasting questions and eliciting probabilities on them. I need your help in deciding, for two given forecasting questions, whether it makes sense to think about their logical combinations/whether it's worth doing so.
For example, we might want to elicit the probability of
'Will Donald Trump win the 2024 US presidential election? AND Will US economic growth exceed 3.5% in 2025?'
because Trump winning the election might potentially (positively or negatively) affect economic growth in the following year.
But we probably wouldn't care about the probability of
'Will Donald Trump win the 2024 US presidential election? AND Will the men's deadlift record be broken in 2025?'
because those seem wholly unrelated.
Can you help me with this? I will just give you two forecasting questions, and you must give me

1. One or more examples of reasons someone might be interested in the logical combination of those questions; based on how realistic these reason(s) are, provide–

2. a score between 0 and 10 to advise me on whether it makes sense to consider their logical combination (with 0 being 'the logical combination is nonsensical, nobody would ever ask something like that', 10 being 'yeah that's a perfectly legitimate question I could imagine seeing that on Manifold or Metaculus')

---

Figure 15: The prompt used to decide whether two questions are related enough to be combined in an instantiated tuple.

---

**Tuple instantiation prompt – OR**

You are a helpful assistant. I will give you two forecasting questions with Yes/No answers. You should then give me the logical OR of these two questions, i.e. the question that would be answered YES if EITHER question is answered YES, and NO otherwise. Notes:

- Your response should be as clear as possible, since the words 'and' and 'or' are used ambiguously in natural language. For example, 'Will P happen or will Q happen? is usually confusing, as it sounds like you are asking which of the two will happen (whereas you're actually seeking a YES/NO answer on whether either of the two will happen). Instead, if there is any chance of confusion, you should give me something like: Will either of the following occur: (a) P (b) Q?

- When the questions allow for a simple rephrasing or factorization (e.g. using words like 'respectively', 'both' or 'either'), go for it.

- If one or both of the given questions is already a logical combination of questions, join them in the most natural way possible. E.g.

  - combine ((P1 OR P2) OR Q) how you would combine (P1 OR P2 OR Q)
  - ((P1 AND P2) OR Q) might have to be combined as something like: Will EITHER of the following occur: (1) BOTH of the following occur: (a) P1 AND (b) P2 (2) Q. Unless a more natural formulation exists.

- Be careful when combining conditional expressions (which often have words like 'given' and 'if'). '(Given A then P) OR (Given B then Q) should be combined as is, rather than messing up the conditions. E.g. a phrasing like 'Will either of the following occur given their respective conditions: (a) Given A then P? (b) Given B then Q?' is good.

- This also applies when only one of the questions is conditional. Like 'P OR (Given A then Q)'should be phrased as something like: 'Will either of the following occur given their respective conditions are met? (a) P (b) Given A, then Q?'.

- Most importantly: make sure you retain ALL the information in the question bodies from BOTH base questions! You cannot discard a single relevant detail. All this is for an experiment to test the logical consistency of forecasters: The combined question you give will be handed to the forecasters without having seen the base questions, so it is critical that all the information in the base questions be included in your logical combination; the resolution criterion for each component should be neatly and clearly provided.

- Also, make sure that the title is self-sufficient independent of the body, i.e. is a question that can be meaningfully answered without looking at the body. So you CANNOT give me a question title like 'Is the following true?' or 'What will happen if the following happens?'

- One type of question you may be given is a single choice from a multiple choice question. For example, you may be given 'Which of these countries will legalize human cloning by 2030? (Japan)'. This is asking if Japan will recognize and legalize human cloning by 2030. Such a question may also itself be a logical combination – e.g. 'Which of these countries will legalize human cloning by 2030? (UK, France, or Germany) is asking if any either of the UK, France, or Germany will legalize human cloning by 2030. Make sure to correctly combine such combinations as previously described.

Figure 16: The prompt used for instantiating OR tuples. We use similar prompts for other checks.

---

**Verification prompt – CONSEQUENCE**

I will provide you with two propositions, P and Q. Your task is to assess whether Q is a proposition that will always be true if P is true. In other words, validate whether Q is a logical implication of P, ensuring that Q will always occur if P is true. Reject if P and Q are completely equivalent. Q should be a logical consequence of P, but not necessarily the other way around. Reject if you need any additional assumptions to derive Q from P. Reject if Q is just formed by making some resolution criteria more vague / not operationalizing them (but accept if it is made by actually loosening some resolution criteria while still precisely defining everything). Reject if Q is 'ERROR: NO CONSEQUENCE FOUND' or something like that.

Example 1:

P: A computer can receive emails.

Q: A computer is connected to the internet.

reasoning: If a computer can receive emails (P), then it must be connected to the internet (Q), as an internet connection is necessary for receiving emails. Therefore, Q is a logical consequence of P.

valid: True

Example 2:

P: The ground is wet.

Q: It is raining.

reasoning: I can easily imagine the ground being wet (P true) without it raining (Q false). So P does not imply Q.

valid: False

Example 3:

P: It is daytime.

Q: The sun has risen and not set yet.

reasoning: The two statements are logically equivalent, as daytime (P) is defined by the sun being above the horizon and not having set yet (Q). So Q is a logical consequence of P, but also completely equivalent to it, therefore not useful to us.

valid: False

Example 4:

P: Will at least 50 percent of the world's population live in Asia by 2050?

Q: Will Asia have at least 3 billion residents by 2050?

reasoning: They probably thought Q was a logical consequence of P because the world population is 8 billion, half of that is 4 billion, so if Asia has more than 4 billion people it must have more than 3 billion people. However, this assumes that the world population in 2050 is 8 billion, which we do not know for certain. Without knowing the world population in 2050, we cannot judge if 50 percent of that is more or less than 3 billion.

valid: False

Example 5:

P: Will ANY of the following happen in 2025? (a) A manned mission to Mars (b) A new Starship launch by SpaceX?

Q: Will a manned mission to Mars happen in 2025?

reasoning: Suppose only a new starship launch happens, but no manned mission to Mars. Then P is true, but Q is false. So Q is not a logical consequence of P.

valid: False

Example 6:

P: Will there be an epidemic of meningococcal diseases in 2025? (An epidemic is defined as a disease with an infection rate of over 15 per 100,000 people over a span of two consecutive weeks.)

Q: Will there be an epidemic of meningococcal diseases in 2025? (An epidemic is defined as a disease that infects a large number of people over a short span of time.)

reasoning: Q just makes the definition of an epidemic more vague, so we do not count it.

valid: False

Example 7:

P: Will there be an epidemic of meningococcal diseases in 2025? (An epidemic is defined as a disease with an infection rate of over 15 per 100,000 people over a span of two consecutive weeks.)

Q: Will there be an epidemic of meningococcal diseases in 2025? (An epidemic is defined as a disease with an infection rate of over 10 per 100,000 people over a span of two consecutive weeks.)

reasoning: Q is a logical consequence of P, as the definition of an epidemic in Q is a subset of the definition of an epidemic in P. If an epidemic is defined as infecting more than 15 per 100,000 people, it must also be true that it infects more than 10 per 100,000 people, as 15 is more than 10.

valid: True

---

Figure 17: The prompt used to verify the generated CONSEQUENCE tuple is logically valid.

## I  FEEDBACK FORM EXAMPLES

---

**Example Entry**

- **title**: Will the United Kingdom deploy an AI-powered public surveillance system with real-time facial recognition capabilities by 2025?

- **body**: Resolution criteria: This question will resolve as Yes if, by December 31, 2025, the United Kingdom has officially deployed a public surveillance system that utilizes artificial intelligence for real-time facial recognition across any part of its public spaces. The system must be capable of identifying individuals and matching them to a database with at least an 85% accuracy rate. The deployment of such a system must be confirmed by an official government announcement or documentation. In the absence of a public announcement, credible reports from at least three major news organizations (e.g., BBC, The Guardian, Reuters) will be considered sufficient evidence. The system must be operational and not in a trial phase. If multiple systems are deployed, the resolution will consider the first system that meets these criteria. In the event of a partial deployment (e.g., limited to specific cities or areas), the question will resolve as Yes if the system is intended to be expanded nationwide. Edge cases, such as temporary deployments for specific events or the use of similar technology in private spaces, will not count towards this question's resolution.

- **resolution_date**: 2025-12-31 00:00:00+00:00

- **metadata**:
    - **tags**: [United Kingdom]
    - **category**: [Artificial Intelligence]

---

**Example Feedback**

- **bad_or_irrelevant_included_information**:

- **unintuitive_or_wrong_resolution_criteria**:

- **too_specific_criteria_or_edge_cases**:

- **ambiguities**: Should specify which public news agencies would count as resolution.

- **edge_cases_not_covered**:

- **general_feedback** :

- **formatting_issues**:

- **rewritten_title:**:

- **rewritten_body**: Resolution criteria: This question will resolve as Yes if, by December 31, 2025, the United Kingdom has officially deployed a public surveillance system that utilizes artificial intelligence for real-time facial recognition across any part of its public spaces. The system must be capable of identifying individuals and matching them to a database with at least an 85% accuracy rate. The deployment of such a system must be confirmed by an official government announcement or documentation. In the absence of a public announcement, credible reports from at least three major news organizations (BBC, The Guardian, Reuters, Bloomberg, New York Times, Washington Post) will be considered sufficient evidence. The system must be operational and not in a trial phase. If multiple systems are deployed, the resolution will consider the first system that meets these criteria. In the event of a partial deployment (e.g., limited to specific cities or areas), the question will resolve as Yes if the system is intended to be expanded nationwide. Edge cases, such as temporary deployments for specific events or the use of similar technology in private spaces, will not count towards this question's resolution.

- **rewritten_resolution_date**:

- **discard_reason**:

---

## J    CONSISTENCY AROUND A QUESTION

There is no particular reason why we need a starting dataset to measure consistency over questions and the corresponding instantiated tuples; a single starting question suffices. We give a preliminary exploration of a pipeline for measuring consistency around a given question. This pipeline is especially useful when we have a dataset of questions and want a consistency metric for each of these questions. For example, to understand how much consistency helps with understanding the correctness of a forecast, we want a *per-question consistency metric* to compare with a dataset of Brier scores.

We follow a similar process as in Section 3.1 and Section 3.2. We start with a dataset of questions we want consistency metrics around, and then few-shot prompt gpt-4o (see Figure 18) to generate related questions for each source question. We follow the deduplication process based on text-embedding-3-small embeddings from OpenAI to ensure diverse questions.

As in Section 3.1, after title creation, we generate question bodies and resolution dates using a few-shot prompt to gpt-4o. Next, this dataset of each source question followed by generated related questions are used to create logical tuples in the same form as in Section 3.1. We ensure that each source question is included in the tuple, along with the necessary number of related questions for the specific check: 1 for NEGATION, 2 for COND, and so on.

For tuples where the order of the questions matter, such as $\text{COND}(P, Q|P, P \wedge Q)$, we allow the source question to take the position of $P$ or $Q$. Overall, we get a dataset of tuples for each source question, such that the source question is included in the tuples. We follow the same steps for verification and evaluation. For evaluation around a source question, we aggregate the consistency metrics by source question.

---

**Synthetic question generation prompt for source question**

Objective: Generate a set of forecasting questions for a forecasting market site like Metaculus or PredictIt. I will provide a source question. Your task is to generate {num_questions} new related questions that are logically related to the provided source question. Each new question should be suitable for probabilistic evaluation and should logically combine with the source question in a meaningful way.
Guidelines:
- The new questions should explore related scenarios, alternate outcomes, consequences and prerequisites of the source question.
- Consider alternate outcomes, timelines, or deeper implications that are connected to the theme of the source question.
- Each question should be binary and can be answered with a probability between 0 and 1.
The source question will optionally include a body (detailed resolution criteria). If the source question has a body, use it to inform the generation of related questions. You still need to generate only single sentences, not detailed resolution criteria.
Examples:
{example_1}
{example_2}
{example_3}
Source question: {source_question}
=¿ Related questions:

---

Figure 18: The prompt used for generating the *title* field of related questions, given a source question.

# K CREATING FQS WITH KNOWN RESOLUTION FROM NEWS ARTICLES

This section describes a pipeline for creating forecasting questions with known ground-truth resolutions using news articles retrieved from NewsAPI. We derive an initial set of forecasting questions directly from the news articles. Then, to ensure broader coverage and mitigate dataset biases inherent to this approach of generating questions, we generate additional questions by spanning their reference classes, modifying key components like location or entity while preserving thematic and temporal consistency.

Finally, we verify and, where necessary, assign ground-truth resolutions to all generated forecasting questions via the Perplexity API (`perplexity/llama-3.1-sonar-huge-128k-online`), see Appendix K.3 The ground truth resolutions given by `perplexity/llama-3.1-sonar-huge-128k-online` are not always correct, but have an error rate of less than 5% when applied to the scraped question dataset.

## K.1 NEWSAPI-BASED FORECASTING QUESTION GENERATION

We use NewsAPI due to its diverse set of sources and free availability, making it suitable for our application. Additionally, we curate a list of reliable news sources, such as Associated Press, which tend to provide more informative and factual content rather than opinion-based articles. These sources yield a higher volume of articles grounded in real-world events that can be effectively transformed into forecasting questions.

We gather daily news articles from 1 July 2024 to 31 August 2024 through NewsAPI. These articles include fields such as the title, content, description, and publication date, and are consolidated into a single file for further processing.

At this stage, we encounter an issue: conflicting news articles from different dates report opposing information. For instance, one article states that President Joe Biden confirms his candidacy for the 2025 U.S. Elections, while a later article claims he withdraws. These discrepancies lead to the generation of forecasting questions with contradictory resolutions.

To address this, we remove older articles that are highly similar to more recent ones by calculating a Named Entity Recognition (NER) similarity score[8], based on the ratio of shared entities to unique ones. Articles surpassing a certain similarity threshold are treated as duplicates, allowing us to discard outdated and repetitive information and resolve the issue as in the Biden problem above.

We feed processed articles to `gpt-4o` to determine their suitability for creating forecasting questions with binary resolutions, judging them based on parameters such as clarity of content, contextual relevance, binary resolution potential, and specificity. The prompt for this is in 19.

---

[8]https://spacy.io/models/en

---

**Example Validated News Article with Reasoning**

- **Article**
  - **Source**
    - *ID:* bloomberg
    - *Name:* Bloomberg
  - **Author:** Bloomberg
  - **Title:** HDFC plans to sell $1.2 billion of loans to plug funds gap
  - **Description:** The bank is in talks with local asset managers including ICICI Prudential AMC, Nippon Life India Asset Management and SBI Funds Management to issue so-called pass through certificates
  - **URL:** https://www.bloomberg.com/news/articles/2024-08-30/hdfc-bank-plans-to-sell-1-2-billion-of-loans-via-rare-debt-tool
  - **Image URL:** https://bl-i.thgim.com/public/todays-paper/tp-news/e3asi7/article68587355.ece/alternates/LANDSCAPE_1200/Private-sector-G29D92OKN.4.jpg.jpg
  - **Published At:** 2024-08-31T13:27:56Z
  - **Content:** HDFC Bank plans to sell as much as 100 billion ($1.2 billion) of loan portfolios using a debt instrument it rarely used, as the nation's banks devise ways to overcome challenges in raising deposits...
- **Validation Result:** true
- **Validation Reasoning:**
  - The article provides clear information about HDFC Bank's plan to sell $1.2 billion of loans, which is a concrete and definitive event.
  - It mentions the involvement of local asset managers, giving context to the transaction.
  - The specificity of the information is sufficient to generate forecasting questions, such as whether the sale will be completed by a certain date or if specific asset managers will participate.
  - The event has a binary resolution potential, as the sale will either occur or not.
  - However, the article's content is truncated, but it still contains enough information to support actionable predictions.
  - Therefore, the article meets most criteria for generating forecasting questions.

---

Articles identified as suitable for forecasting questions are then processed by our *Rough Forecasting Question Generator* module using gpt-4o. This generator follows structured guidelines (described in 20) to extract clear and unambiguous YES/NO questions based solely on the article's information. Each question consists of a clear and precise title that adheres to temporal guidelines, ensuring the resolution date aligns with the article's month. The body provides essential context without superfluous details, and the ground-truth resolution is directly derived from the source article.

Further, we include a *pose date* (set to October 1st, 2023) in the prompt to ensure temporal clarity. This is only relevant for NewsAPI-based FQs and should not be confused with the created_date in Appendix C.1. For example, when an event is referenced as happening in 2024, the *pose date* prompts the LLM to add relevant context, preventing disambiguation issues for forecasters unfamiliar with the event. The resulting intermediate data structure, containing the question's title, body, and resolution, is then passed to the *Final Forecasting Question Generator* for further refinement.

---

**Example Rough FQ Data**

- **Article Title:** Death toll is now 8 in listeria outbreak tied to Boar's Head deli meat, CDC says
- **Article Description:** It's the largest listeria outbreak since 2011. On July 29, the recall was expanded to include all foods produced at the firm's plant in Jarratt, Virginia.
- **Article Content:** At least eight people have died after being infected with listeria from Boar's Head deli meats tied to a massive recall last month, federal health officials said Wednesday. The new food poisoning to... [+7300 chars]
- **Article URL:** https://apnews.com/article/listeria-boars-head-recall-d57985525441b6c5dffd310769b0e6c5
- **Article Published At:** 2024-08-28T21:15:00Z
- **Forecasting Question Title:** Will the listeria outbreak tied to Boar's Head deli meat result in more than 5 confirmed deaths by August 2024?
- **Forecasting Question Body:**
    - This question resolves as YES if, by August 31, 2024, there are official reports confirming more than 5 deaths attributed to the listeria outbreak linked to Boar's Head deli meats.
    - Official confirmation must come from credible sources such as the CDC or equivalent health authorities, and reported by at least two reputable news outlets.
    - If the death toll remains 5 or fewer, the question resolves as NO.
- **Forecasting Question Resolution:** true

---

Our experiments indicate that `claude-3.5-sonnet` produces better-phrased questions than `gpt-4o`; however, it occasionally generates hallucinated content and introduces fabricated details not found in the original article. To leverage Claude's strengths in phrasing while addressing this concern, we incorporate a validation prompt into the *Final Forecasting Question Generator* process. This prompt (21) assesses the intermediate (rough) forecasting questions on multiple criteria, ensuring clarity and removing elements that suggest a direct derivation from a news article, including the article's publication date. After validating these questions, we rephrase them to minimize overly specific details, thereby enhancing their generality and facilitating their predictability.

The *Final Forecasting Question Generator* subsequently validates the resolutions of the rephrased forecasting questions (using 22). This process involves prompting `gpt-4o` to evaluate the generated questions against their respective source news articles. The LLM determines whether a binary resolution is applicable or if the question cannot be answered based on the information provided in the article. This approach effectively filters out questions that do not derive directly from the news articles and imposes the necessary constraints of clarity and specificity. By focusing solely on the factual content available at the time of publication, the generator ensures that the resolutions are both definitive and accurate. We then verify the NewsAPI-generated FQs with a common FQ verification step to ensure correct structure and format.

We generate a dataset of forecasting questions using NewsAPI articles published between July 1, 2024, and August 31, 2024, inclusive, as described in the above pipeline.

---

**Example Final FQ**

- **ID:** 43b7f07f-02e2-432c-8912-1311aa5f1af8
- **Title:** Will Hawaii enact legislation restricting the public carrying of non-firearm weapons by August 2024?
- **Body:** This question will resolve as YES if, by August 31, 2024, Hawaii officially passes and enacts legislation that imposes new restrictions on the public carrying of non-firearm weapons, such as bladed weapons or other non-firearm implements previously affected by the recent legal change. The legislation must specifically address the carrying of these weapons in public spaces. For a YES resolution, the new law must be officially enacted and reported by at least two reputable news sources (e.g., Associated Press, Reuters, local Hawaiian news outlets). If no such legislation is passed and enacted by the specified date, or if any enacted legislation does not specifically restrict the public carrying of non-firearm weapons, the question will resolve as NO.
- **Resolution Date:** 2024-08-31T23:59:59
- **Question Type:** binary
- **Data Source:** synthetic
- **Created Date:** 2024-06-30T23:59:59
- **URL:** None
- **Metadata:**
  - **Article Information:**
    * **Article URL:** https://apnews.com/article/hawaii-gun-rights-weapons-second-amendment-f61c972ebbb28fb21baa28385fa069cd
    * **Article Date:** 2024-08-28 10:46:38
    * **Article Description:** Second Amendment activists in Hawaii are celebrating a recent legal change that allows them to carry not just guns but other weapons — from battle-axes to butterfly knives — openly in public. Hawaii has long had strict weapons laws and some of the lowest rate...
    * **Article Title:** Bikinis, surfboards and battle-axes? Hawaii loosens long-strict weapons laws after court ruling...
    * **Article Content:** HONOLULU (AP) Hawaii's tourist hotspot of Waikiki is known for bikinis, shopping and surfboards. But resident Andrew Roberts has recently introduced a different item on evening walks through his neighborhood... [+5086 chars]
  - **Pose Date:** 2023-10-01 00:00:00
- **Resolution:** false

---

### K.2 GENERATING DIVERSE FQS THROUGH REFERENCE CLASS SPANNING

A critical issue in forecasting inquiries is the inherent bias towards *current phenomena*, which results in an overrepresentation of outcomes associated with actively reported events. For instance, if a forecasting question posits whether Colorado will conduct a referendum on abortion rights by July 2024 and the answer resolves as *Yes* due to media coverage, this introduces a distortion within the dataset. Similar inquiries—such as whether Nevada will pursue a comparable referendum or whether Colorado will address unrelated topics like gaming regulation—may be inadequately represented or entirely omitted, thus perpetuating a bias towards current phenomena. This imbalance prevents us from effectively testing forecasters' ability to predict a wider array of potential scenarios, limiting the evaluation to outcomes associated with current events and reported phenomena.

To mitigate this bias, we advocate for the implementation of the *Reference Class Spanner* methodology, which utilizes gpt-4o to systematically create a set of additional forecasting questions within the same reference class [9] by modifying essential entities or components (prompted with 23). This approach ensures that the dataset reflects a more extensive spectrum of outcomes rather than being disproportionately skewed towards events reported as occurring.

The *Reference Class Spanner* method generates new forecasting questions by varying one to two core components of the original question while preserving its resolution date and thematic structure, thereby facilitating broader scenario exploration. For example, it transforms the question "Will

---

[9]https://en.wikipedia.org/w/index.php?title=Reference_class_problem&oldid=1229577621

Tesla complete a major software upgrade for over 1.5 million vehicles in China by August 2024?" into "Will Ford complete a major software upgrade for over 1.5 million vehicles in the states by August 2024?" This approach promotes diversity in potential outcomes and significantly mitigates bias toward positive outcomes by producing a set of high-quality forecasting questions within the same reference class. By prompting the LLM to change multiple key components simultaneously—such as the company name or location—we ensure that the questions generated remain plausible and relevant. We verify the structure of the generated questions and subsequently input them into our *Perplexity Verification Module* to attach ground truth resolutions.

Table 8: NewsAPI Generated FQs. Represents the number of data points generated until creation of reference spanned FQs using K.2.

| Data | July 2024 | August 2024 | Total |
|------|-----------|-------------|-------|
| Initial News Articles | 533 | 486 | 1019 |
| Validated News Articles | 381 | 363 | 744 |
| Rough FQ Data | 457 | 375 | 832 |
| Final Validated FQs | 117 | 104 | 221 |
| Reference Spanned FQs | 2517 | 2246 | 4763 |

> **Examples of reference spanned FQs**
>
> - **Original Question**
>   - ID: 54667f62-5119-4c3e-bedf-37e3b94bd49f
>   - Title: Will India report a successful winter crop season for wheat and rapeseed by August 2024?
>   - Body: This question will resolve as YES if, by August 31, 2024, India reports a successful winter crop season for wheat and rapeseed, characterized by yields meeting or exceeding the average of the past five years. The success must be confirmed by official agricultural statistics from the Indian Ministry of Agriculture and Farmers' Welfare or at least three reputable news sources (such as Reuters, Bloomberg, or The Economic Times). For this question, 'successful' is defined as the combined production of wheat and rapeseed being at least 5% above the five-year average. If the winter crop season does not meet these criteria, or if insufficient data is available to make a determination, the question resolves as NO.
> - **Spanned Questions**
>   - Spanned Question 1
>     * ID: 041133ab-2358-4c06-9580-86ade14f4026
>     * Title: Will Pakistan report a successful winter crop season for wheat and sugarcane by August 2024?
>     * Body: This question will resolve as YES if, by August 31, 2024, Pakistan reports a successful winter crop season for wheat and sugarcane, characterized by yields meeting or exceeding the average of the past five years. The success must be confirmed by official agricultural statistics from the Pakistan Ministry of National Food Security & Research or at least three reputable news sources (such as Reuters, Bloomberg, or The Economic Times). For this question, 'successful' is defined as the combined production of wheat and sugarcane being at least 5% above the five-year average. If the winter crop season does not meet these criteria, or if insufficient data is available to make a determination, the question resolves as NO.
>   - Spanned Question 2
>     * ID: 42c713c2-ecea-4208-876d-af0b38dab566
>     * Title: Will Turkey report a successful winter crop season for wheat and hazelnuts by August 2024?
>     * Body: This question will resolve as YES if, by August 31, 2024,...
>   - Spanned Question 3
>     * ID: bbe55403-c062-44cf-a0a8-2d96e68d9f2a
>     * Title: Will Iran report a successful winter crop season for wheat and pistachios by August 2024?
>     * Body: This question will resolve as YES if, by August 31, 2024,...

### K.3 VERIFYING THE FQ RESOLUTIONS USING A PERPLEXITY-BASED QUESTION RESOLVER

To ensure a high-quality benchmark, we verify or attach resolutions to every forecasting question generated in the previous stages. This verification process uses the Perplexity API (`llama-3.1-sonar-huge-128k-online`), querying models with internet access to determine if the question can be resolved with current information. If the question is resolvable, we obtain and attach the resolution. In cases where Perplexity cannot resolve the question, or if the resolution differs from the one originally derived from the source article, we discard that question.

For questions formed through reference class spanning, we directly attach the resolution obtained from Perplexity. For those generated from news articles, we focus on verifying the accuracy of the initial resolutions to ensure consistency and reliability in our dataset. As of the creation of the NewsAPI FQ dataset up until K.2, Perplexity maintains an accuracy of over 95%, with half of the discrepancies arising due to contradictory internet data (which makes the resolution unclear even to the authors). Due to the potential of such label noise, we adopt the Brier score instead of the log scoring rule for all ground truth metrics.

We create a ground-truth resolved dataset (`20240701_20240831_gpt-4o_spanned_resolved.jsonl`) comprising of 2621 forecasting questions which is used for tuple instantiation. Further, we filter out 1000 questions (`20240701_20240831.jsonl`) from this set, consisting of all of the NewsAPI

Table 9: Question Verification and Resolution Data for July and August 2024. Notably, the final count of resolved questions is lower than the combined totals for both months, as questions with existing resolutions that differ from those suggested by Perplexity are discarded.

| Data | July 2024 | August 2024 | Total |
|---|---|---|---|
| Total Questions Generated | 2517 | 2246 | 4763 |
| Filtered for Verification | 2516 | 2246 | 4762 |
| Questions Discarded After Perplexity | 1005 | 1090 | 2095 |
| Resolved with Final Resolution Attached | 1511 | 1156 | 2667 |
| | **Final Total Questions Resolved** | | 2621 |

generated FQs and a subset of the reference-spanned questions, to use as a ground-truth dataset in our experiments.

**News Article Validation Prompt**

**System Prompt:**

- You are an AI agent responsible for evaluating news articles to determine their suitability for generating forecasting (prediction) questions that can be answered with a definitive YES or NO. Assess each article against the following criteria to ensure clarity, relevance, and factual accuracy:

  - **Clarity of Content**: Is the information presented clearly and straightforwardly? Reject articles that are overly convoluted or difficult to understand.
  - **Focus on Definitive Events**: Does the article discuss concrete events that have occurred or are planned? Evaluate articles referencing past events based on their clarity and context.
  - **Contextual Relevance**: Does the article provide adequate context for the events discussed? While some background gaps are acceptable, the article should allow for a reasonable understanding of the events.
  - **Specificity of Information**: Is the information detailed enough to formulate precise forecasting questions? Reject articles that are too vague to support clear predictions.
  - **Binary Resolution Potential**: Does the article imply a resolution that can be confirmed as TRUE (YES) or FALSE (NO)? Articles may contain subjective elements but should lead to a binary outcome.
  - **Completeness of Information**: Does the article provide sufficient detail to create multiple high-quality forecasting questions? Brief articles are acceptable as long as they contain enough information.
  - **Numerical Clarity**: If applicable, does the article present clear thresholds or metrics for numerical data? Some ambiguity is acceptable, but numerical references should be understandable.
  - **Sufficiency for Definitive Resolution**: Does the article provide enough information to formulate forecasting questions that yield definitive resolutions from the current date until the specified resolution date in {*month_name*}, {*year*}? Ensure the content supports actionable predictions based on concrete events, assuming the current date is {*pose_date*}.
  - **Truncated Information**: Truncated information is NOT a cause for rejection. Accept articles that can form prediction questions, even if they reference past events not covered by the LLM's knowledge.

- An article that meets most of these criteria is considered "complete" and suitable for generating forecasting questions, even if it contains minor ambiguities or references past events that may not be fully known.

**User Prompt:**

- Please evaluate the following news article based on the established criteria for completeness: {*source_news_article*}

- Based on your assessment, determine if the article is "complete" and suitable for generating forecasting questions. Provide a brief justification for your decision.

Figure 19: Validation prompt used to judge whether a processed news article can be used to create a forecasting question with a binary resolution.

---

**Rough FQ Generation Prompt**

**System Prompt:**

- **Objective:** Generate forecasting questions that can be definitively answered with YES or NO, based on the provided news articles, while testing a forecaster set in the past.

- **Forecaster's Context:** The forecaster's present date is set to {*pose_date*}, so all questions must be framed as if this is the current date. Although the articles may reference future events, your questions must be phrased in a way that the forecaster cannot detect the actual date of question creation.

- **Clarity & Precision:**
  - Each question must be clear, specific, and unambiguous.
  - Avoid subjective terms like "significant" or any similar ambiguity.
  - Do not reference sensitive topics such as religion, politics, or gender.

- **No Temporal Hints:**
  - Do **not** include any information or context that implies the question was created after {*pose_date*}.
  - Ensure no indication that the article is used to inform the question, keeping the creation date fully hidden.

- **Resolution Period:**
  - If you phrase the resolution date as "by {*month_name*}, {*year*}", then resolution of each question must remain definitive and applicable from the current date until {*month_name*}, {*year*}.
  - If you phrase the resolution date as "in {*month_name*}, {*year*}", then resolution of each question must remain definitive and applicable for the month of {*month_name*} in {*year*}.
  - Ensure the question's outcome is verifiable and binary (YES or NO) during this period.

- **Context from Articles:**
  - Use concrete events from the articles, providing enough background to make the question understandable.
  - Ensure questions are diverse, covering a wide range of topics without bias or triviality.

- **Goal:** Generate a diverse set of precise and objective forecasting questions that seamlessly align with the forecaster's assumed timeline without revealing the true creation date or source of the information.

Figure 20: Prompt used to generate an intermediate (rough) forecasting question consisting of just the title, body and resolution from a news article. Continued on the next page.

---

**Rough FQ Generation Prompt (Continued)**

**User Prompt:**

- **Task:** Based on the provided news article, generate multiple high quality forecasting questions that follow these structured guidelines. Each question must consist of a title, body, and resolution. The generated forecasting questions must only be formed using information from the article and no other extrapolations or inferred information.

- **News Article:** {*source_article*}

- **Title Guidelines:**

  – YES/NO Clarity: Formulate each question so that it can be definitively answered with a YES or NO, based on the article's content.

  – Avoid Sensitive Topics: Do not reference religion, politics, gender, or race.

  – Direct and Precise: Titles must be straightforward and unambiguous, avoiding vague terms.

  – Resolution Date: Include a resolution date using the format "by {*month_name*}, {*year*}?" or "in {*month_name*}, {*year*}?", whichever is more suitable for the context.

  – Context for Clarity: Provide enough context if event names may not be clear as of the forecaster's present date ({*pose_date*}).

  – Named Entities: There is no limit on the number of named entities from the article, but the question should avoid becoming overly specific.

  – Planned or Announced Events: Frame planned events as proposals or announcements rather than completed facts, including sufficient context to avoid ambiguity.

- **Body Guidelines:**

  – Disambiguation: Stay focused on the title's core question without introducing unrelated details.

  – No Extra Information: Only include relevant context to support the title.

- **Resolution Guidelines:**

  – Binary Outcome: Resolutions must be clearly marked as True for YES and False for NO.

  – Stable Outcome: Ensure the resolution remains consistent and unchangeable until the resolution date.

  – Definitiveness: The resolution must be verifiable based solely on the content of the article.

- **General Guidelines:**

  – Avoid Specific Knowledge: Do not require specialized knowledge that could disadvantage forecasters unfamiliar with niche topics.

  – Base Questions on Article Content: Ensure all forecasting questions are directly derived from the article's content, avoiding speculative or inferred details.

Examples included in the prompt have been skipped for brevity.

---

**Final FQ Validation and Rephrasing Prompt**

**System Prompt:**

- You are an expert in validating and rephrasing forecasting (prediction) questions based on news articles. A forecasting question consists of a title, body, and resolution.

- Your task is to ensure that each question adheres to the established guidelines and to enhance the phrasing of valid questions. It is important to note that while we are formulating these questions after knowing the resolutions, the forecaster will assume they are answering them as of {*pose_date*}. The resolution date for the questions should be set as {*month_name*}, {*year*}.

- Guidelines to be followed are:

    1. Forecaster's Context:
        - The forecaster's present date is set to {pose_date} so all questions must be framed as if this is the current date. Although the articles may reference future events, your questions must be phrased in a way that the forecaster *cannot* detect the actual date of question creation.

    2. Clarity & Precision:
        - Each question must be clear, specific, and unambiguous.
        - Avoid subjective terms like "significant" or any similar ambiguity.
        - Do not reference sensitive topics from religion, politics, or gender.

    3. No Temporal Hints:
        - Do *not* include any information or context that implies the question was created after {pose_date}.
        - Ensure no indication that the article is used to inform the question, keeping the creation date fully hidden.

    4. Resolution Period:
        - If you phrase the resolution date as "by {*month_name*}, {*year*}", then resolution of each question must remain definitive and applicable from the current date until {*month_name*}, {*year*}.
        - If you phrase the resolution date as "in {*month_name*}, {*year*}", then resolution of each question must remain definitive and applicable for the month of {*month_name*} in {*year*}.
        - Ensure the question's outcome is verifiable and binary (YES or NO) during this period.

    5. Factual Basis:
        - Questions should be directly supported by the article content and not include fabricated information.

**User Prompt:**

- You are tasked with the following steps:

    1. Validation:
        - Check if the forecasting question adheres to the provided guidelines. A question is valid if it aligns with the guidelines.

    2. Rejection:
        - Reject the question if it violates any guidelines. The rejected form should be: {*example_rejected_fq*}.

    3. Rephrasing:
        - For valid questions, rephrase them to enhance clarity, specificity, and compliance with the guidelines while retaining the original intent. Do NOT add any new information that wasn't included in the original question.

- High-Quality Forecasting Question Examples: *Some Examples*

- Task:

    - Carefully validate and rephrase the following forecasting question: {*source_rough_fq_data*}.

---

Figure 21: Prompt used validate the structure of NewsAPI generated forecasting questions and then rephrase them to enhance predictability.

---

**Final FQ Resolution Validation Prompt**

**System Prompt:**

- You are an AI agent tasked with verifying the resolution of forecasting questions based solely on the content of a provided news article. Your role is crucial in ensuring that the resolutions are definitive and accurately reflect the information available at the time the question was posed.

    – Factual Basis: The resolution should be based on the factual information present in the news article.

    – Publication Perspective: Your assessment should be made from the perspective of the article's publication date, not any other date.

    – Inference Guidelines: Reasonable inferences are acceptable, but do not fabricate details or speculate beyond what is stated in the article.

    – Response Options: Use the 'None' option if there is absolutely no information in the article that allows for a reasonable inference of either YES or NO. If the article provides any relevant context or information that can lead to a definitive answer, choose either 'True' or 'False'.

**User Prompt:**

- Consider the following news article:

    – Title: {*article_title*}

    – Description: {*article_description*}

    – Content: {*article_content*}

    – Date: {*article_date*}

- Now, consider this forecasting question: {*question_title*}

- For additional context, use the following information to disambiguate the question: {*question_body*}

- Your task is to determine the resolution of the question based solely on the factual information present in the news article, assuming the article's publication date is the current date. Return:

    – 'True' if the answer to the question can be reasonably inferred as YES.

    – 'False' if the answer to the question can be reasonably inferred as NO.

    – 'None' if there is absolutely no information in the article that allows for a reasonable inference of either YES or NO.

- Please provide a brief justification for your answer, citing specific details from the article that support your reasoning.

Figure 22: Prompt used to verify whether a forecasting question formed using NewsAPI has the correct resolution using the source article.

---

**Forecasting Question Generation Prompt**

**System Prompt:**

- **Objective:** Generate high-quality forecasting questions (FQs) by spanning the reference class of a given source question. Your goal is to enhance the diversity of the dataset while minimizing bias.

- **Reference Class:** In probability theory, a reference class refers to a group of similar events or outcomes that share common features. Your task is to create new forecasting questions by varying key components (e.g., location, topic, action, or subject) of the source question, ensuring they stay within the same reference class.

- **Key Requirements:**
  - Consistency in structure and thematic integrity with the original source question.
  - Vary only one to two key elements while ensuring logical consistency.
  - The new questions should remain unresolved.
  - Use the same resolution date as the source question.

- **Question Structure:**
  - YES/NO clarity, avoid sensitive topics, direct and precise titles.
  - Context for clarity with a clear binary outcome for resolutions.
  - Retain the same resolution date as the source forecasting question.

**User Prompt:**

- The source forecasting question is: {*source_forecasting_question*}.

- **Instructions:**
  - Identify the core components (event type, location, key subjects, or outcomes) of the source question.
  - Replace one to two significant elements with a similar entity while maintaining logical structure.
  - Ensure balance and neutrality, with a diverse probability distribution of possible outcomes.
  - Verify that the new questions remain realistic, relevant, and unresolved as of now.
  - Create {*num_questions*} forecasting questions by spanning the reference class of the provided source question.

Figure 23: Prompt used to generate high-quality forecasting questions by varying key elements of a source question using reference class spanning. Examples have been omitted for brevity.

## L  2028 SYNTHETIC QUESTIONS CONSISTENCY CHECK DATASET

This section presents a set of questions with a resolution date in 2028. These questions were created using a prompt similar to the one in Figure 14, with two key additions:

1. **Target Resolution Date**: The prompt specifies a target resolution date, in this case January 1, 2028. And asks the model to propose questions about events happening before the resolution date, or in the year of the resolution date. About half of the initial few shot examples are modified with the chosen resolution date.

2. **Creation Date**: The prompt includes a creation date, in this case October 1, 2024. This is crucial to prevent the generation of questions that could be trivially answered on the creation date, but are in the future from the perspective of the model knowledge cutoff.

Below are two example questions from this dataset:

---

**Examples of Synthetic Questions with 2028 Resolution**

- **Synthetic Question 1**
  - ID: 2f2e7e08-5241-40ba-8ad1-5a037408388c
  - Title: Will Australia's GDP grow by at least 3% annually for three consecutive years before January 1, 2028?
  - Body: This question will be resolved as 'Yes' if Australia's GDP, as reported by the Australian Bureau of Statistics, grows by at least 3% annually for three consecutive years at any point between October 1, 2024, and January 1, 2028. The growth rate must be based on official annual GDP growth figures released by the Australian Bureau of Statistics.
  - Additional Details:
    * Question Type: Binary
    * Resolution Date: 2028-01-01 00:00:00
    * Created Date: 2024-10-01 00:00:00
    * Data Source: Synthetic
    * Category: Economy & Business
    * Tags: Australia
- **Synthetic Question 2**
  - ID: 93eafe80-e854-4d29-bbe7-da52d851025c
  - Title: Will Switzerland hold a national referendum on joining the European Union before January 1, 2028?
  - Body: This question will be resolved as 'Yes' if, between the creation date of this question (October 1, 2024) and January 1, 2028, Switzerland holds a national referendum on the issue of joining the European Union. The referendum must be officially sanctioned by the Swiss government and the results must be publicly announced.
  - Additional Details:
    * Question Type: Binary
    * Resolution Date: 2028-01-01 00:00:00
    * Created Date: 2024-10-01 00:00:00
    * Data Source: Synthetic
    * Category: Geopolitics
    * Tags: Switzerland

---

We create 900 verified (see verification paragraph in Section 3.1) base forecasting questions resolving in 2028. From these, we run the consistency check instantiation pipeline in Section 3.2, to create 300 tuples per check, for a total of 3000 tuples. We then run a single forecaster on this benchmark to get a sense of baseline performance on our dataset.

The consistency metrics on this dataset provide the best proxy available for comparing general long-term forecasting ability of LLM forecasters, but many caveats apply.

Future work may consider creating a similar benchmark with a secret subset, to prevent new forecasters from being trained to cheat on this benchmark. Note that, due to the lack of ground truth

Table 10: Consistency metrics for CoT-GPT-4o-08 on the synthetic 2028 questions dataset.

| Check | Arbitrage | | Arbitrage Scaled | | Frequentist | |
|---|---|---|---|---|---|---|
| | Avg | Frac | Avg | Frac | Avg | Frac |
| NEGATION | 0.033 | 49% | 0.016 | 49% | 0.178 | 50% |
| PARAPHRASE | 0.014 | 37% | 0.007 | 37% | 0.107 | 38% |
| CONDCOND | 0.044 | 65% | 0.011 | 54% | 0.296 | 89% |
| EXPEVIDENCE | 0.031 | 35% | 0.008 | 23% | 0.186 | 50% |
| CONSEQUENCE | 0.003 | 7% | 0.001 | 7% | 0.021 | 8% |
| AND | 0.020 | 23% | 0.007 | 18% | 0.080 | 25% |
| OR | 0.016 | 36% | 0.006 | 24% | 0.105 | 37% |
| ANDOR | 0.034 | 46% | 0.008 | 36% | 0.190 | 49% |
| BUT | 0.050 | 58% | 0.017 | 54% | 0.317 | 81% |
| COND | 0.042 | 66% | 0.014 | 60% | 0.242 | 71% |
| Aggregated | 0.029 | - | 0.010 | - | 0.172 | - |

resolutions, accidental training on the dataset does not automatically invalidate any consistency metric, as opposed to what happens with standard benchmarks.

## REPRODUCIBILITY STATEMENT

We include the questions, forecasting results, and consistency results necessary to reproduce all tables and plots in the paper. The forecasts data [10] is organized by forecaster, with two directories for each forecaster:

1. Ground truth forecasting results:
   - JSONL file, where each entry has (question, boolean resolution, forecast, per-question Brier score, metadata and reasoning traces)
   - JSON file: total Brier score, calibration, other metrics
2. Consistency checks results:
   - JSONL file with raw results, where each entry (questions, forecasts, consistency violations, metadata and reasoning traces)
   - JSON file: summary statistics (e.g., average violation)

The consistency check results directories have a substring `tuples` in the directory name. For the 2028 synthetic dataset, we have only the consistency check result directories.

---

[10]https://github.com/dpaleka/consistency-forecasting/tree/b093e5134f219ca4d82720bb996ec1fb850024ae/src/data/forecasts

