# OpenReview forum: "Consistency Checks for Language Model Forecasters"
_ICLR.cc/2025/Conference — ICLR 2025 Oral_

### Official Review · Reviewer_QioK · 2024-10-26

**Soundness:** 4
**Presentation:** 4
**Contribution:** 4
**Rating:** 8
**Confidence:** 4

**Summary:**

The paper proposes to evaluate large language model (LLM) forecasters by assessing the consistency of their predictions through a "consistency check", since ground truth is often unavailable about the future – we have to live and see!). Thus, the paper proposes an arbitrage-based and a frequentist metrics to detect logical inconsistencies. It finds good correlations between these consistency metrics and forecasting performance metrics (such as Brier score) tested on events that have already happened from a datasets established by this paper (one from prediction markets and one synthetically generated from news articles).

**Strengths:**

TLDR: This paper takes on a massive challenge and delivers as much as imaginable in such a hard setting.

**S1. Novel evaluation approach with a lot of effort and care to do things right:** The authors cleverly circumvent the need for ground truth in long-term forecasts by developing consistency metrics, validated by correlation measures and by carefully gathering datasets with events that resolved “between May 1, 2024, and August 15, 2024.”, so after the knowledge cutoff of the models. The consistency checks are clever and make predictions more interpretable, since they appeal to logic. Also, the paper includes many and diverse LLMs into testing. Well done!

**S2. Useful artefacts:** The paper creates a long-horizon consistency benchmark for forecasts resolving in 2028. This adds a massive a practical tool for future benchmarking, especially of more advanced LLMs that are yet to come.

**S3. Lots of details for reproducibility:** The paper provides a solid appendix with comprehensive documentation, including forecasting question structures, pipeline details, and explicit consistency conditions. This enhances reproducibility, makes the paper experimental design and choices easy to follow, and can make the potential applicability across related forecasting tasks real, by enabling other researchers to implement this on their own domain.

**S4. The paper’s writing is excellent.** Otherworldly.

**Weaknesses:**

**Minor:** Limited breadth of data domains: Dependency on NewsAPI and Manifold and Metaculus for some forecast questions, while practical, may limit the breadth of testing topics and introduce data biases. But this is a minor issue, since the challenge taken on by this paper is massive, and the experiments were conducted with a lot of care, and the methodology presented in this paper can be extended to as many domains as possible (S3).

**Questions:**

Since the correlation between consistency metrics and forecasting performance metrics is conducted on events that have already concluded, is it not possible that the LLMs already know that, thus they are more accurate and more consistent in their predictions? I would assume that you chose events that resolved “between May 1, 2024, and August 15, 2024.” (L203), because that would be after the knowledge cutoff, right. But are we absolutely 100 percent sure that we know the actual knowledge cutoff of these models (since OpenAI loves switching things behind the scenes in the API without letting users know?)

L134, the phrase "We set V(F(x1), . . . F(xn)) to be the maximum such 'minimum profit'" could benefit from a rephrasing – "We define V(F(x1), . . . F(xn)) as the maximum achievable minimum profit from the arbitrageur…"

---

> ### Author Response · Authors · 2024-11-23
>
> We thank the reviewer for recognizing our evaluation approach, research artifacts, and writing!
>
> > Limited breadth of data domains: Dependency on NewsAPI and Manifold and Metaculus for some forecast questions, while practical, may limit the breadth of testing topics and introduce data biases
>
> We expand the NewsAPI-derived dataset by spanning reference classes of questions. This expands the distribution of topics, but does not resolve all biases. This is described in Appendix J.2.
> We will release our code to make it easy to create consistency evaluations on any dataset of base forecasting questions. We defer creating better forecasting question datasets for future work.
>
> >  I would assume that you chose events that resolved "between May 1, 2024, and August 15, 2024." (L203), because that would be after the knowledge cutoff, right. But are we absolutely 100 percent sure that we know the actual knowledge cutoff of these models (since OpenAI loves switching things behind the scenes in the API without letting users know?)
>
> That is true; we picked data after the stated cutoff date for all the models. In cursory inspections of our data, we did not observe forecasts of 0% and 100% probability that would be proof of data leakage.
>
> We additionally thank the reviewer for noting the phrasing issue, which we have now fixed in the updated submission.

---

> > ### Comment · Reviewer_QioK · 2024-11-24
> > **Response by rev**
> >
> > Thank you for your answer. I would like to keep my original scoring of the paper (8, accept).

---

### Official Review · Reviewer_YSMD · 2024-10-30

**Soundness:** 2
**Presentation:** 2
**Contribution:** 3
**Rating:** 5
**Confidence:** 4

**Summary:**

This paper studies forecasting binary events using LLMs. They make numerous contributions involving consistency violations of binary forecasts. They introduce consistency metrics, propose an evaluation pipeline including the creation of two forecasting datasets, analyze correlation between consistency violations and the quality of the forecasts as gauged by the Brier scoring rule, a proposed forecaster to reduce violations, and a new benchmark where events are resolved in 2028.

**Strengths:**

This paper explores the important subject of using LLMs to forecast future events. Clearly a lot of effort has gone into the work, from the data generation aspects formalizing and running the evaluation pipeline.

Another strength of the paper, at least from what I understand, is that the authors have tried to avoid data contamination issues that are prevalent in the literature involving the use of LLMs. They have also created a new benchmark for events resolving in 2028, which could be beneficial to the community.

**Weaknesses:**

I’m a bit confused by the focus of the study itself, which is to use logical consistency to evaluate a forecaster before resolution of the events themselves; presumably, the point of this is to evaluate forecasts sooner. But what is the value of using consistency as an early marker? There are likely confounding factors at play here that determine a model's affect on both forecasting consistency and the performance on the task. For instance, maybe a certain class of models (perhaps larger or trained in certain ways) are better forecasters – and many of these features are observable anyway, so why not just use these instead of consistency? I don’t understand why the authors have gone down this path and would like to know more about what motivated this line of work. Also, the observed correlation in the two instances doesn’t provide too much evidence for the claims, in my view, so I wonder about the generalizability of the claims. Frankly, I do not fully see the merits yet but I’m open to changing my mind.

In my opinion, way too much (important) information is in the Appendix. I appreciate the space limitations, but I feel a lot of important material is missing from the main text, such as details about the frequentist metrics and details about the datasets. I did not get a chance to look closely at various parts of the Appendix.

The metrics themselves seem overly and possibly unnecessarily complicated. The rationale for these were also not provided clearly. What about something simpler, such as distances between scores that involve S? Perhaps I’m not understanding the complexity of scoring when arbitrary predicates are used for consistency.

The literature review seems too sparse. For instance, there are several papers on consistency in general, and probably many others on forecasting with LLMs. I will not point them out specifically but it is relatively easy to find more work. Perhaps the authors can consider expanding the related work section.

Section 5 came across as somewhat rushed, perhaps as it is somewhat preliminary and the results are not too promising, from what I understand.

**Questions:**

Here are some further questions and suggestions:

How large are the datasets created? Could the authors elaborate? I probably missed this information.

How can one be assured that the data contamination issue is not a problem for forecasting? I understand the authors have mentioned that they try to build a dataset that avoids this issue through time stamps – could they confirm if they are sure that there is no data contamination, in that the training data does not directly include information about the forecasted events?

I noted several typos and related issues in the manuscript. Here are a few (most after resolution):

Line 82: “analyses can extend to …”

Line 107: should Forecast be in [0,1] rather than equal to the range? I’m not sure about this one.

Line 161 (footnote 1): “Manifold Markets use …”

Line 148-152: This choice seems arbitrary and merely pointing to an example with the sum of probabilities being 1.1 is not helping.

Line 169: “individual step …”

Line 242: what is “sensical”? Do you mean “sensible”?

Figure 1: The description mentions 3 metrics. Should there be 3 panels? Also, there is a space before a period in the caption.

Figure 2: What is COND? What is CONDCOND? It seems like some information is missing.

Line 324: “a proxy …”

Note that there are some citation style issues in Section 6.

---

> ### Author Response · Authors · 2024-11-23
>
> We thank the reviewer for recognizing that our work explores an important topic in a high-effort way. Below, we answer the reviewer’s main questions.
>
> > For instance, maybe a certain class of models (perhaps larger or trained in    certain ways) are better forecasters – and many of these features are observable anyway, so why not just use these instead of consistency?
>
> A good forecaster, intuitively, requires (1) strong reasoning; (2)  lots of knowledge about the world (either in weights or retrieval); and (3) a robust and consistent way of converting the above into probabilities. There are benchmarks for (1) and (2), but not that many for (3). In addition, we think  here is value in doing an outside view on evaluation, instead of relying on measurements of different skills that we think estimate the forecasting accuracy correctly.
>
> > The metrics themselves seem overly and possibly unnecessarily complicated. The rationale for these were also not provided clearly. What about something simpler, such as distances between scores that involve S? Perhaps I’m not understanding the complexity of scoring when arbitrary predicates are used for consistency.
>
> Thank you for the comment; the rationale for the exact metrics we use is indeed not explained well within the main body. We will update the paper before publication. The key reason why we expend effort to do metrics properly is *principled aggregation and comparison of inconsistency scores*.
>
> In the standard Bayesian setting, one’s beliefs are consistent or not: there either is a Dutch book (a way to bet against the forecaster’s beliefs to get infinite profit) or there is not. In practice, a forecaster’s beliefs (even on the same question) are never perfectly consistent across runs. If an election model has a presidential candidate at 48% with one random seed and 50% on the other, this is not a reason to discard it as completely flawed. If we discarded every single Dutch-bookable forecast, even very good forecasters might be discarded as inconsistent.
>
> A good metric should have answers to the below:
>
> - Is a Negation forecast tuple of (P, not P) \= (0.5, 0.6) better or worse than a Negation forecast tuple of (P, not P) \= (0.89, 0.01)?  (We think the latter is much worse in many applications, such as forecasting tail risk.)
> - Is a Negation forecast tuple of (P, not P) \= (0.5, 0.6) better or worse than a Cond forecast tuple of (P, P | Q, P and Q) \= (0.2, 0.5, 0.2)?
> - How to compute a single-figure inconsistency score over a batch of questions, without certain logical checks (or probability regions) unfairly dominating the score?
>
> We do not exactly understand what “distances between scores that involve S” means. Prior work (Fluri et al.) just came up with some metric for each score that normalizes to \[0, 1\]. There are two major issues with this approach:
> (1) Most simple metrics will be linear and fail the intuition on the (0.5, 0.6) vs (0.89, 0.01) example above.
> (2) It’s unclear how to compare and aggregate scores from different consistency checks.
>
> To tackle these issues, our approach ensures that all consistency scores share a common “unit”. For example, in the arbitrage metric, to aggregate inconsistencies, we sum up the *profit* made by an arbitrageur across questions.
>
>
>
> > I'm a bit confused by the focus of the study itself, which is to use logical consistency to evaluate a forecaster before resolution of the events themselves; presumably, the point of this is to evaluate forecasts sooner. But what is the value of using consistency as an early marker?
>
> There is no way to directly measure anything about a forecaster’s performance over future events. We either need to hope for generalization from backtesting on past events, or we need *some* metrics to apply to unresolved predictions.
>
> > The literature review seems too sparse. For instance, there are several papers on consistency in general, and probably many others on forecasting with LLMs.
>
> Thank you for the advice\! We improved the related work section now with several more papers. The modifications are in red. We welcome suggestions on other papers that we should comment on.
>
> > How large are the datasets created? Could the authors elaborate? I probably missed this information.
>
> This is in Section 3.3: “We run each of these forecasters on 5000 tuples in total (for each of the 10 checks, we use 200 tuples from scraped questions and 300 from NewsAPI questions), ...”.
>
> We additionally thank the reviewer for noting the notation issues and typos, which we have now fixed.

---

> > ### Comment · Reviewer_YSMD · 2024-11-26
> > **Thank you**
> >
> > I thank the authors for providing several clarifications in their response. I acknowledge the immense effort put into this work but remain skeptical about various facets, as mentioned in my review. I will be open to further appreciating the merits of the work during subsequent discussions.

---

### Official Review · Reviewer_N1zy · 2024-11-04

**Soundness:** 4
**Presentation:** 3
**Contribution:** 3
**Rating:** 8
**Confidence:** 3

**Summary:**

The paper introduces a novel approach to evaluating language model (LLM) forecasters, given the challenge of evaluating predictions with unknown future outcomes. The authors propose a consistency check framework that uses arbitrage and hypothesis testing as a metric, measuring logical inconsistencies across different, related forecasting questions. They build an automated system for generating forecasting questions, applying consistency metrics, and demonstrate that consistency scores correlate well with forecasting accuracy as measured by Brier scores.

**Strengths:**

* The manuscript is well-written, and the ideas are easy to follow.
* The paper effectively addresses the challenge of evaluating LLM forecasters by proposing new metrics based on arbitrage and hypothesis testing.
* It provides a thorough derivation and demonstrates a statistically significant correlation between the metrics proposed and Brier score.
* The authors also curated a dataset of forecasting questions, enhancing the evaluation for the future works.

**Weaknesses:**

* The ArbitrageForecaster section would benefit from additional practical details, including parameter tuning, framework specifications, and methods for applying adjustments across various LLM models. Adding a framework visualization could further enhance clarity.

**Questions:**

* I wonder what's the performance for o1-mini for the negation and paraphrase experiment in Figure 3? Or is there any reason why the authors decided to evalute GPT4o instead of o1-mini?
* Given that the long-horizon benchmark evaluates consistency without ground truth, how do authors see the questions without ground truth being indicative of forecasting ability? Are there specific applications where such consistency checks alone could be considered sufficient?

---

> ### Author Response · Authors · 2024-11-23
>
> We thank the reviewer for recognizing our thorough evaluation, metrics, and writing.
>
> Here, we answer the questions about the choices we made in our experiments with ArbitrageForecaster.
>
> > I wonder what's the performance for o1-mini for the negation and paraphrase experiment in Figure 3?
>
> Our estimates of cost constraints and iterating on the ArbitrageForecaster with smaller models were the only factor here, as detailed in Appendix F1. Internally, we might have overestimated the costs a bit, but it is still not a trivial cost. A single call to ArbitrageForecaster with even just depth=r, arbitraging checks=\[Neg, Paraphrase\] requires 3^r LLM calls; measuring a single consistency check of each type requires \~21\*3^r calls. Assuming 600 input tokens and 600 output tokens (our rough estimate of the expected hidden reasoning length) and performing the experiment on 200 questions, this amounts to a total token cost of \~400million tokens, or about $3000 with o1-mini for this single plot.
>
> **Design choices for ArbitrageForecaster.** Appendix F.1 goes into the detailed theoretical motivations for the specific set-ups we experimented with. In summary:
>
> (1) we hypothesized that ArbitrageForecaster will be particularly effective on “symmetric” and “deterministic” checks; thus we studied ArbitrageForecaster instantiated with Neg.
>
> (2) we hypothesized that there would be consistency gains from increasing depth, thus we studied recursive ArbitrageForecaster setups.
>
> (3) We were interested in knowing if the consistency gain after arbitraging on a single check would persist after applying further checks. Thus we studied the case of \[Neg,Paraphrase\] applied together.
>
> (4) We hypothesized that ExpectedEvidence may improve ground truth and consistency across the board.
>
> Apart from that, the limited scope of these experiments was due to significant cost barriers to branching on multiple checks simultaneously. Future research could focus on a cost-effective way to implement ArbitrageForecaster: for example, in training, one may consider randomly sampling some subset of checks to arbitrage on for each question.

---

> > ### Comment · Reviewer_N1zy · 2024-11-24
> >
> > Thank you for the clarifications. I would like to maintain my rating for accepting the paper.

---

### Official Review · Reviewer_7grX · 2024-11-04

**Soundness:** 3
**Presentation:** 4
**Contribution:** 3
**Rating:** 8
**Confidence:** 2

**Summary:**

The authors expand on forecasting consistency checks pioneered as part of Fluri et al. 2024 (Evaluating Superhuman Models with
Consistency Checks). While that paper presented a proof-of-concept limited set of consistency checks on a single model, this work concretizes these checks into a set of usable metrics and demonstrates some of these metrics' high correlation with forecasting performance (as measured by Brier score).

These metrics include an arbitrage metric, incurring penalties according to how much guaranteed profit an arbitrageur can receive by exploiting the inconsistencies in the probabilities if they were market prices, and a frequentist metric, which uses a hypothesis testing setup to test whether a given forecaster could be sampling from a distribution based on adding noise to a consistent world model. The authors conduct experiments using a new already-resolved test dataset, and release another long-term forecasting dataset to resolve in several years (2028).

**Strengths:**

- Strong communication of the problem and proposed resolution
- Detailed data generation for both already-resolved experiments and an open-ended future forecasting benchmark
- Extensive discussion of decisions & limitations
- Principled theory and evaluation.
- Correlation plots indicate that the consistency scoring criteria could be used to generally select better forecasters.
- Theoretical and practical method to improve consistency at evaluation time.

**Weaknesses:**

1. Test-time ArbitrageForecaster does not gain consistency in an emergent set of checks; only those it is specifically designed to address. This reduces the impact of the paper slightly.
2. New consistency metrics are rather easily Goodharted; one can imagine training a model on a vast set of synthetic forecasting problems respecting consistency checks in order to gain nearly perfect consistency scores with no improved knowledge of the future. This would limit the long-term efficacy of these metrics as a way to distinguish quality forecasters, unless novel checks are often added (which seems increasingly difficult without dramatically increasing the complexity of new checks).
3. No comparison with system from Halawi et al. 2024: "Unfortunately, we could not reproduce the system from Halawi et al. (2024) at the time of writing, due to deprecations in the Google News API (we could not obtain access to the alternative Newscatcher API)." It is unclear if this is addressable in the longer term. It seems this comparison would be useful.

Nitpicks (grammar etc.):

- 2536 "with over the discrepancies"
- 2924 " are modifies "
- 184 May want to use P, R here for the input set of questions to distinguish from the output tuple P, Q and the input set of questions P, Q.

**Questions:**

1. Can you report confidence interval for the Brier score of a new model based on its consistency scores (so long as it is not Goodharted on consistency?) - e.g. with Fisher's z transformation?
2. Can one report, in addition to the correlation between Brier and each consistency metric, a ranking metric on the models' Brier performance based on the consistency ranking? It seems that selecting the best model/few models based on consistency scores is a critical use case.
3. Is there any way to ensure contamination-free or low-contamination consistency checking? Can we design an arbitrary search space for consistency checks that is large enough to not be saturated trivially by consistency training (see weakness 2).

---

> ### Author Response · Authors · 2024-11-23
>
> We thank the reviewer for recognizing our ideas and theory as principled, as well as commending the writing and the execution.
>
> **Goodharting and ArbitrageForecaster.** The ArbitrageForecaster set-ups we tested do not empirically improve ground truth evaluation or consistency on other checks  — this suggests that consistency scoring could indeed be Goodharted. However, note that most of our experiments with ArbitrageForecaster only involved arbitraging on a *single* check (apart from one experiment with both Negation & Paraphrase). We avoided adding more checks due to the large cost of experiments. It’s easy to imagine how a bad forecaster could Goodhart a single check — simply reporting the same probability for all questions will pass ParaphraseChecker and ExpectedEvidenceChecker; reporting prob=0.5 for all questions will further pass the Negation check — but we expect that being consistent under a variety of checks becomes increasingly difficult without a consistent world model.
>
> > Is there any way to ensure contamination-free or low-contamination consistency checking? Can we design an arbitrary search space for consistency checks that is large enough to not be saturated trivially by consistency training (see weakness 2).
>
> Our intuition is that checks with multiple source questions do not seem to be easily saturated by consistency training, as we can always sample new questions to combine into tuples, and the forecaster does not know which questions are in the tuple when answering any particular question.
>
> **Other**. Due to the GNews API changes described in the paper, we are not at this point able to fully reproduce the Halawi et al. forecasters. We believe it is likely that better LLM forecasters will be built in the near future, and hope that these will then be evaluated on our future-based consistency benchmark. To this end, we will release our code, and will strive to make it easy for future LLM forecasting papers to check consistency on our 2028 events benchmark.

---

> > ### Comment · Reviewer_7grX · 2024-11-29
> >
> > Thanks for addressing my comments! I am satisfied and maintain my score.

---

### Meta-Review · Area_Chair_pW9a · 2024-12-21

**Metareview:**

This paper addresses the challenge of evaluating language model (LLM) forecasters for future events, where ground truth is inherently unavailable at the time of prediction. The authors propose a novel approach using "consistency checks" based on logical relationships between forecasting questions, introducing two key metrics: an arbitrage-based metric and a frequentist metric. These metrics measure inconsistencies in LLM predictions, such as predicting contradictory outcomes (e.g., both Democrats and Republicans winning an election with high probability). The paper demonstrates a strong correlation between these consistency metrics and traditional forecasting performance metrics (e.g., Brier score) on a dataset of resolved events. Additionally, the authors release a long-term consistency benchmark for events resolving in 2028, providing a valuable tool for future research. The work includes a comprehensive evaluation across multiple LLMs and a detailed discussion of methodology and limitations.

#### Contribution
1. **Novel Consistency Metrics**: The paper introduces an arbitrage-based metric and a frequentist metric for evaluating LLM forecasters' consistency, providing a way to assess forecasting quality without ground truth.
2. **Empirical Validation**: Demonstrates a strong correlation between the proposed consistency metrics and traditional forecasting performance metrics (Brier score) on resolved events, validating the approach.
3. **Long-Term Benchmark**: Releases a consistency benchmark for events resolving in 2028, offering a practical tool for future benchmarking of advanced LLMs.
4. **Methodological Rigor**: Provides a detailed methodology, including data generation, pipeline design, and consistency conditions, enhancing reproducibility and applicability.
5. **Comprehensive Evaluation**: Conducts experiments across a diverse set of LLMs, ensuring broad applicability and robustness of the findings.

#### Weaknesses
1. **Potential for Goodharting**: The proposed consistency metrics could be manipulated by training models to pass the checks without improving actual forecasting ability, especially if the checks are limited in scope.
2. **Data Contamination Concerns**: While the authors take steps to avoid data contamination (e.g., using events resolved after LLMs' knowledge cutoff), there is no absolute certainty that the models are contamination-free, particularly given opaque API changes by model providers.
3. **Limited Data Domains**: The reliance on specific data sources (NewsAPI, Manifold, Metaculus) may introduce biases and limit the breadth of tested topics, potentially affecting the generalizability of the findings.
4. **ArbitrageForecaster Limitations**: The ArbitrageForecaster improves consistency only on specific checks it is designed to address, potentially limiting its broader applicability.

**Additional Comments On Reviewer Discussion:**

1. **Goodharting and ArbitrageForecaster (7grX):**
   - **Concern**: The consistency metrics could be easily Goodharted, limiting their long-term efficacy.
   - **Response**: The authors acknowledged that Goodharting is possible, particularly with single-check setups, but argued that consistency across multiple checks becomes increasingly difficult without a robust world model. They suggested that checks involving multiple source questions are harder to saturate through training, as the forecaster cannot anticipate which questions will be combined.

2. **Data Contamination (QioK):**
   - **Concern**: Events used for evaluation might be within the LLMs' knowledge, compromising the validity of the results.
   - **Response**: The authors confirmed that they selected events resolved after the stated knowledge cutoff for all tested models. They did not observe clear signs of data leakage (e.g., 0% or 100% predictions) and emphasized the challenge of absolute certainty given proprietary model updates.

3. **Limited Data Domains (QioK):**
   - **Concern**: Reliance on specific data sources (NewsAPI, Manifold, Metaculus) may introduce biases.
   - **Response**: The authors explained that they expanded the dataset by spanning reference classes of questions to broaden topic coverage. They also noted that their methodology is extensible to other domains and plan to release code facilitating broader application.

4. **ArbitrageForecaster Details (N1zy):**
   - **Concern**: Lack of practical details on ArbitrageForecaster implementation, including parameter tuning and model application.
   - **Response**: The authors provided cost estimates for running ArbitrageForecaster experiments, highlighting the high computational cost as a limiting factor. They also outlined theoretical motivations for their experimental setups, such as focusing on symmetric and deterministic checks.

5. **Metric Complexity and Rationale (YSMD):**
   - **Concern**: The proposed metrics seem overly complex without clear rationale.
   - **Response**: The authors clarified that the complexity is necessary for principled aggregation and comparison of inconsistency scores. They emphasized the need for metrics to handle nuanced inconsistencies (e.g., distinguishing between minor and major violations) and to aggregate scores across different checks fairly.

6. **Value of Consistency as an Early Marker (YSMD):**
   - **Concern**: The value of using consistency as an early marker of forecasting ability is unclear, especially given potential confounding factors.
   - **Response**: The authors argued that consistency checks provide an indirect measure of forecasting quality when ground truth is unavailable, complementing other benchmarks that test reasoning and knowledge. They emphasized the importance of an "outside view" on evaluation.

#### Final Decision Weighting
The authors' responses effectively addressed the reviewers' concerns, particularly around the rationale for metric design and the potential for Goodharting. The paper's contributions, including the novel consistency metrics and the long-term benchmark, are significant and well-validated through empirical correlation with traditional forecasting metrics. While limitations such as potential data contamination and domain bias remain, these are acknowledged and mitigated to a reasonable extent. The authors should ensure that the final version includes the promised clarifications and improvements, particularly regarding the rationale for metric design and the extensibility of their methodology across domains.

---

### Decision · Program_Chairs · 2025-01-22

Accept (Oral)